# JDM: JOINT DISTRIBUTION MODELING FOR FINE-GRAINED TEXT-TO-VIDEO GENERATION

## ABSTRACT

Text-to-video (T2V) generation enables AI systems to create videos from textual descriptions, with applications in entertainment, education, and content creation. Recent advances in video diffusion models have improved visual quality, yet they struggle with fine-grained text-video alignment, often leading to attribute mismatches, incorrect object interactions, and compositional failures. In this paper, we identify that this limitation stem from a predominant focus on video reconstruction rather than explicitly learning structured text-video correspondences. To address this, we propose Joint Distribution Modeling (JDM), a novel framework that enhances fine-grained alignment by modeling the joint distribution of video content and object masks. Unlike prior methods that rely on external constraints, JDM inherently learns structured mappings between textual descriptions and video regions, improving compositional consistency. We theoretically demonstrate that JDM improves text-video alignment by directly optimizing for fine-grained correspondences rather than relying on implicit learning from data. Experimental results show that JDM significantly enhances alignment while maintaining high video quality. Furthermore, JDM unifies video generation and segmentation within a single framework, paving the way for more structured and controllable text-to-video synthesis.

## 1    INTRODUCTION

Diffusion models (Ho et al., 2020; Song et al., 2022; Karras et al., 2022) and flow matching methods (Lipman et al., 2023; Liu et al., 2022) have substantially advanced video generation capabilities (Ho et al., 2022b; Singer et al., 2022; Chen et al., 2023a; 2024; Ho et al., 2022a; Wang et al., 2023a; Kondratyuk et al.; Zhou et al., 2023; Blattmann et al.; Wang et al., 2023c; Zhang et al., 2023a; Ruan et al., 2024; Guo et al., 2023b; Chefer et al., 2025). Nevertheless, accurately aligning textual descriptions with generated video content remains challenging, particularly when texts contain complex compositional structures (Liu et al., 2023; Tian et al., 2024; Feng et al., 2025; Huang et al., 2024; Sun et al., 2025; Wang et al., 2023b; 2024b). For instance, as illustrated in Figure 1(a), the attribute "yellow," associated with "curtain", erroneously leak onto the sofa, indicating a failure to correctly bind attributes to corresponding objects. Similarly, Figure 1(b) demonstrates another form of semantic leakage, wherein the action "swim," contextually linked to a fish, incorrectly propagates to a horse. These examples highlight the existing models' difficulty in comprehending fine-grained semantic relationships, underscoring the need for enhanced methods capable of capturing and maintaining precise text-video correspondences.

Essentially, text-to-video generation aims to learn the conditional probability density function $p(\mathbf{x}_0 \mid \mathbf{y})$, where $\mathbf{y}$ represents the textual input and $\mathbf{x}_0$ denotes the video. In conventional training paradigms (Ho et al., 2020; Song et al., 2022; Lipman et al., 2023; Karras et al., 2022), the model explicitly regresses the noise added to the video rather than directly enforcing text-to-video alignment. As a result, any correspondence between text and video emerges implicitly from the training data, with the network autonomously determining how to leverage the conditioning text $\mathbf{y}$. However, there is only **global correspondence** between the text and the video in the training dataset, without any **fine-grained correspondence** (i.e., which part of the text corresponds to specific regions of the video). This lack of fine-grained correspondence makes it difficult for the model to generate videos that accurately reflect compositional and complex textual descriptions.

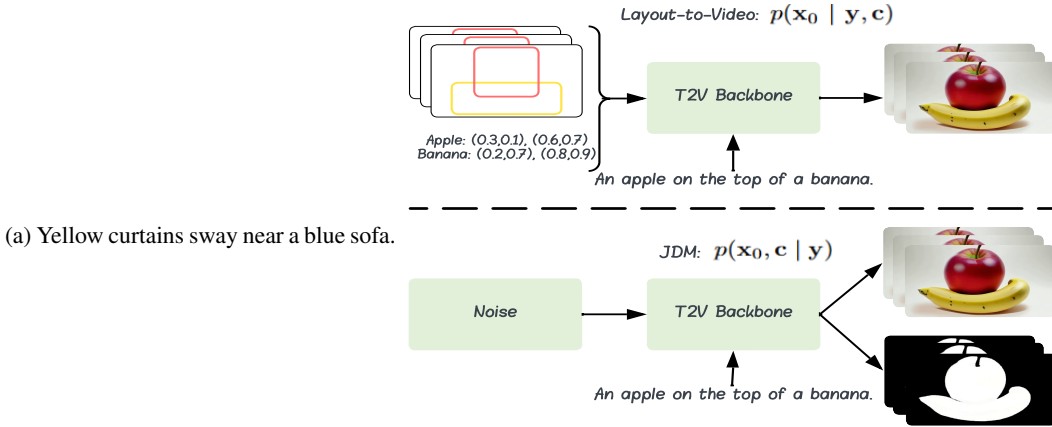

(a) Yellow curtains sway near a blue sofa.

(b) A fish swims gracefully in a tank as a horse gallops outside.

(c) Layout-to-Video models $p(\mathbf{x}_0 \mid \mathbf{y}, \mathbf{c})$, and require additional signals $\mathbf{c}$ during inference. JDM models $p(\mathbf{x}_0, \mathbf{c} \mid \mathbf{y})$, start with pure noise, while can generate both the video and its mask.

Figure 1: (a) & (b): Existing T2V model failed to understand fine-grained text-video correspondence and accurately generate the video content. (c): Difference between Layout-to-Video and JDM.

Existing methods (Feng et al., 2025; Huang et al., 2024; Lian et al., 2024; Wang et al., 2024a; Xie et al., 2023; Li et al., 2023b; Huang et al., 2023a; Guo et al., 2023a; Zi et al., 2024) address the challenge of achieving fine-grained text-video alignment by introducing auxiliary conditioning signals during inference. These methods typically model the conditional distribution $p(\mathbf{x}_0 \mid \mathbf{y}, \mathbf{c})$ , where $\mathbf{c}$ represents auxiliary signals such as pixel-level layouts (Feng et al., 2025; Lv et al., 2023), bounding boxes (Wang et al., 2024a), or spatial coordinates (Lian et al., 2024; Wang et al., 2024c) generated by external models. While incorporating these signals constrains the original text-to-video generation process by limiting the space of $p(\mathbf{x}_0 \mid \mathbf{y})$, this approach does not inherently enhance the model's fundamental understanding of textual semantics. Consequently, the generation will still fail if the model lacks accurate semantic comprehension from the outset. Furthermore, reliance on these supplementary conditions often leads to significant increases in model complexity, and computational overhead, and such signals may not always be practically obtainable, particularly when generating novel or previously unseen content.

In this paper, we propose a novel approach to enhance fine-grained text–video alignment without introducing additional signals during inference. We begin by investigating the limitations of standard diffusion and flow matching models in achieving fine-grained alignment (Sec. 3.1). Building on this analysis, we introduce *Joint Distribution Modeling* (JDM), which incorporates fine-grained correspondence by modeling the joint distribution $p(\mathbf{x}_0, \mathbf{c} \mid \mathbf{y})$. Rather than relying on external constraints, JDM learns a structured latent space where textual descriptions naturally map to corresponding visual regions (Sec. 3.2). Specifically, we leverage object masks paired with their respective regional descriptions as fine-grained correspondence signals. In contrast to existing layout-to-video models, JDM improves text–video alignment with only minimal additional parameters. Moreover, as illustrated in Figure 1, JDM requires only text as input during inference while simultaneously generating both the video and its corresponding mask.

To validate the effectiveness and generalizability of our approach, we implement JDM on two distinct video generation models: a DiT-based model (CogVideoX-2B (Hong et al., 2022)) and a U-Net-based model (ModelScopeT2V (Wang et al., 2023a)). Extensive experimental results demonstrate that our method significantly enhances fine-grained text–video alignment while preserving high visual quality. Furthermore, our work highlights the potential of leveraging generative models for both video generation and video segmentation.

## 2 RELATED WORKS

**Compositional Text-to-Video Generation**: While current video generation models can synthesize videos from simple text prompts, they often struggle when generating videos with multiple objects

or following complex instructions (Zhang et al., 2023b; Mo et al., 2023; Ma et al., 2023; Qin et al., 2023; Choi et al., 2023; Avrahami et al., 2023). This challenge arises from the need to compose objects with diverse temporal and spatial relationships. Vico (Yang & Wang, 2024) regularizes the attention maps associated with each token to improve the translation from each of the text token to video content. VideoTetris (Tian et al., 2024) introduces a spatial-temporal composition mechanism for handling compositional changes in long videos. Several prior works (Feng et al., 2025; Lin et al., 2023; Lian et al., 2024; Huang et al., 2024; Chen et al., 2023b; Feng et al., 2023; Hou et al., 2024) address this issue by planning a layout based on text prompts and integrating this layout into video generation. However, such approaches require layout information during inference and do not inherently improve the model's text-video alignment. For instance, GenMAC (Huang et al., 2024) leverages multiple MLLMs with iterative redesign and regeneration. **Different from layout-to-video generation methods, JDM enhances compositional Text-to-Video generation by improving the model's intrinsic capabilities without requiring additional signals during inference, making it complementary to existing methods such as layout-to-video generation. This design enables seamless integration with other compositional approaches for further performance improvements.**

**Unified Video Generation.** Another closely related line of research is unified video generation, which uses the same backbone and unified latent space to simultaneously generate videos alongside additional auxiliary signals. These auxiliary signals—such as optical flow, dense segmentation maps, and depth maps—provide discriminative supervision during training, forcing the model to develop better visual understanding and generation capabilities. For instance, UniGS (Qi et al., 2023) incorporates mask inpainting and entity segmentation, demonstrating the superiority of such unified generation approaches. VideoJAM predicts optical flow alongside video frames to enhance motion generation quality. UDPDiff (Yang et al., 2025) incorporates video depth and segmentation maps, enabling the model to perform multiple tasks while achieving improved video generation performance. Most recently, WorldWeaver (Liu et al., 2025) demonstrates that jointly outputting depth maps and RGB videos makes the model more 3D-aware, helping to address the drifting issue in long video generation. **Our proposed framework, JDM, differs from these approaches in several key aspects. First, unlike unified video generation models that predict dense scene-level signals (e.g., depth, optical flow) to enhance overall quality, JDM models the joint distribution of regional masks and video conditioned on regional text descriptions. This regional modeling approach directly addresses fine-grained correspondence between text and visual regions, which dense scene-level signals cannot capture. Second, through our theoretical derivation, we show that modeling regional correspondence enables the model to better understand fine-grained compositional relationships.**

## 3 METHODS

Diffusion models (Ho et al., 2020; Song et al., 2021; 2022; Song & Ermon, 2020) approximate real-world data distributions by learning a series of transformations from a simple, known distribution (e.g., Gaussian) to the target data distribution. In particular, diffusion models define a forward stochastic differential equation (SDE) of the form:

$$d\mathbf{x}_t = f(\mathbf{x}_t, t)\, dt + g(t)\, d\mathbf{w}_t,$$

where $f(\mathbf{x}_t, t)$ is the *drift coefficient* that governs the deterministic evolution of the state $\mathbf{x}_t$ over time, $g(t)$ is a time-dependent diffusion coefficient that scales the stochastic component, and $d\mathbf{w}_t$ denotes an increment of standard Brownian motion. To generate new samples, diffusion models solve the corresponding reverse SDE:

$$d\mathbf{x}_t = \left[ f(\mathbf{x}_t, t) - g(t)^2 \nabla_{\mathbf{x}_t} \log p_t(\mathbf{x}_t) \right] dt + g(t)\, d\overline{\mathbf{w}}_t,$$

where $\nabla_{\mathbf{x}_t} \log p_t(\mathbf{x}_t)$ represents the *score function* at time $t$, and $d\overline{\mathbf{w}}_t$ is a Brownian motion term defined in the reverse time direction.

### 3.1 WHY TEXT-TO-VIDEO DIFFUSION MODELS STRUGGLE WITH COMPOSITIONAL TEXT

When training a conditional generative model (e.g., for text-to-video synthesis), the goal is to approximate the conditional score function $\nabla_{\mathbf{x}_t} \log p_t(\mathbf{x}_t \mid \mathbf{y})$, where $\mathbf{y}$ represents the conditioning variable (such as text). A straightforward approach would be to train a conditional network

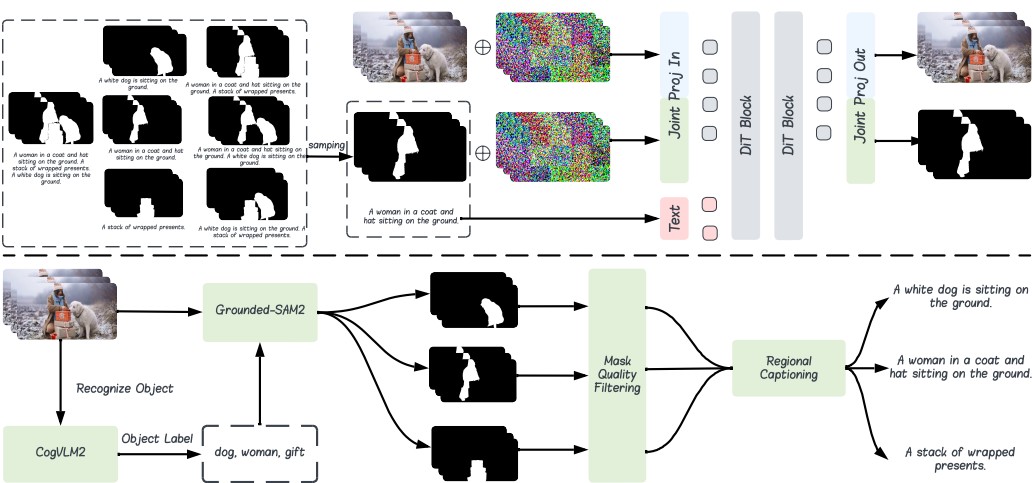

Figure 2: **Top:** JDM training pipeline. Masks and their corresponding regional captions are randomly sampled from all combinations of regional masks and texts. Different combinations are sampled across training epochs, allowing the model to focus on varying video regions and regional texts, thereby enabling fine-grained correspondence learning. The model is trained to jointly denoise both masks and video frames, effectively modeling the joint distribution $p(\mathbf{x}_0, \mathbf{m}_0 \mid \mathbf{y})$. **Bottom:** Overview of the data preprocessing pipeline.

$s_\theta(\mathbf{x}_t, t, \mathbf{y})$ using the following loss function:

$$\mathcal{L}_{\text{naive}} = \mathbb{E}_{(\mathbf{x}_t, \mathbf{y}), t} \left[ \lambda(t) \left\| s_\theta(\mathbf{x}_t, t, \mathbf{y}) - \nabla_{\mathbf{x}_t} \log p_t(\mathbf{x}_t \mid \mathbf{y}) \right\|_2^2 \right].$$

where $\lambda(t)$ is a positive weighting function for score matching loss at different timesteps (Kingma et al., 2023; Song et al., 2021). However, the conditional probability density function $p_t(\mathbf{x}_t \mid \mathbf{y})$ is intractable in general. Diffusion models (Ho et al., 2020; Song et al., 2021; Song & Ermon, 2020) bypass this challenge by conditioning on a known point $\mathbf{x}_0$ and marginalizing over its distribution $p(\mathbf{x}_0 \mid \mathbf{y})$, which can be expressed as:

$$p_t(\mathbf{x}_t \mid \mathbf{y}) = \int p_t(\mathbf{x}_t \mid \mathbf{x}_0, \mathbf{y}) \, p(\mathbf{x}_0 \mid \mathbf{y}) \, d\mathbf{x}_0. \tag{1}$$

In the unconditional case (i.e., without $\mathbf{y}$), the density $p(\mathbf{x}_t \mid \mathbf{x}_0)$ is tractable and can be directly utilized for training. However, in conditional case, we further assume that the perturbation kernel is independent of $\mathbf{y}$ given $\mathbf{x}_0$[1]:

$$p_t(\mathbf{x}_t \mid \mathbf{x}_0, \mathbf{y}) = p_t(\mathbf{x}_t \mid \mathbf{x}_0) = \mathcal{N}(\sqrt{\bar{\alpha}_t}\mathbf{x}_0, (1 - \bar{\alpha}_t)\boldsymbol{I}). \tag{2}$$

where $\bar{\alpha}_t$ are schedule parameters for diffusion. This simplifies the loss to (Full derivation in Appendix):

$$\mathbb{E}_{(\mathbf{x}_0, \mathbf{y}), t, \mathbf{x}_t} \left[ \lambda(t) \left\| s_\theta(\mathbf{x}_t, t, \mathbf{y}) - \nabla_{\mathbf{x}_t} \log p_t(\mathbf{x}_t \mid \mathbf{x}_0) \right\|_2^2 \right].$$

With $\boldsymbol{\epsilon}$-parameterization, diffusion loss is defined as follows:

$$\mathcal{L}_{\text{diff}} = \mathbb{E}_{(\mathbf{x}_0, \mathbf{y}), t, \boldsymbol{\epsilon}} \left[ \left\| \boldsymbol{\epsilon}_\theta(\mathbf{x}_t, t, \mathbf{y}) - \boldsymbol{\epsilon} \right\|_2^2 \right], \tag{3}$$

The primary issue resides in Equation 2, where the perturbation kernel is made independent of $\mathbf{y}$ given $\mathbf{x}_0$. This simplifies the process and makes $p(\mathbf{x}_t \mid \mathbf{x}_0, \mathbf{y})$ tractable. However, this assumption results in the noise being independent of $\mathbf{y}$ given $\mathbf{x}_0$. Consequently, the sole connection between $\mathbf{x}$ and $\mathbf{y}$ arises from the fact that the loss is computed on paired samples $(\mathbf{x}_0, \mathbf{y})$ sampled from the training dataset. It is important to note that the text-video pair $(\mathbf{x}_0, \mathbf{y})$ represents a global relationship, and no fine-grained correspondence is established (for instance, which part of the text

---

[1]It is worthy noting, even if $\mathbf{x}_0, \mathbf{y}$ are paired data, $p_t(\mathbf{x}_t \mid \mathbf{x}_0, \mathbf{y})$ and $p_t(\mathbf{x}_t \mid \mathbf{x}_0)$ are different in general.

corresponds to which region of the video). *In other words, the standard training paradigm only enforces a global text-video relationship and does not require the model to infer detailed, fine-grained correspondences.*[2] As a result, when the model is prompted with complex and compositional text, it is unreasonable to expect that the model will understand each individual concept and generate the corresponding content in the video accurately.

### 3.2 JDM: Joint Distribution Modeling

To enhance the alignment between textual and visual modalities, we aim to incorporate fine-grained text-video correspondence during training, moving beyond reliance on global correspondences. Recognizing that objects serve as the fundamental concepts in video, we utilize object masks along with their associated regional descriptions as the basis for fine-grained text-video correspondence. We begin by defining the following concepts:

- $\mathbf{m}^i$: The object mask $\mathbf{m}^i$ for the $i$-th object in the video, following diffusion notation, denote $\mathbf{m}_0^i$ as the clean mask at diffusion timestep 0.
- $\mathbf{y}^i$: The regional description $\mathbf{y}^i$ associated with the video region masked by $\mathbf{m}^i$.
- $\mathbf{y}$: The caption corresponding to the entire video, obtained by concatenating all regional descriptions $\mathbf{y}^i$, i.e., $\mathbf{y} = \text{concat}(\mathbf{y}^1, \mathbf{y}^2, \ldots, \mathbf{y}^n)$.

In addition, we define a mask oracle $\Theta(\mathbf{m}^i \mid \mathbf{x}, \mathbf{y}^i)$, represented as a probability density function, which generates the mask $\mathbf{m}^i$ given $\mathbf{x}$ and $\mathbf{y}^i$:

$$\mathbf{m}^i \sim \Theta(\mathbf{m}^i \mid \mathbf{x}, \mathbf{y}^i) \tag{4}$$

Our objective is to incorporate the fine-grained text-video correspondence into video generation (represented by the oracle $\Theta(\mathbf{m}^i \mid \mathbf{x}, \mathbf{y}^i)$), thereby enabling the model to infer the correspondence between the regional description $\mathbf{y}^i$ and the specific region $\mathbf{m}^i$ in the video $\mathbf{x}$. Simultaneously, we aim to preserve the generative capability of the network by accurately modeling $p(\mathbf{x}_t \mid \mathbf{y})$. To accomplish these dual goals, we train the network to approximate the joint score function of the video and its corresponding mask, conditioned on the regional description. Formally, we enforce that

$$s_\theta(\mathbf{x}_t, \mathbf{m}_t^i, t, \mathbf{y}^i) \approx \nabla_{\mathbf{x}_t} \log p(\mathbf{x}_t, \mathbf{m}_t^i \mid \mathbf{y}^i). \tag{5}$$

**Modeling of the Oracle** $\Theta(\mathbf{m}^i \mid \mathbf{x}, \mathbf{y}^i)$: For the joint distribution $p(\mathbf{x}_t, \mathbf{m}_t^i \mid \mathbf{y}^i)$, we adopt the following factorization:

$$p(\mathbf{x}_t, \mathbf{m}_t^i \mid \mathbf{y}^i) = p(\mathbf{x}_t \mid \mathbf{y}^i) \, \Theta(\mathbf{m}_t^i \mid \mathbf{x}_t, \mathbf{y}^i). \tag{6}$$

In other words, we decompose the joint distribution into the product of the conditional probability density function $p(\mathbf{x}_t \mid \mathbf{y}^i)$ and the mask oracle $\Theta(\mathbf{m}_t^i \mid \mathbf{x}_t, \mathbf{y}^i)$ as defined in Equation 4 (since the mask is determinable given $\mathbf{x}_t$ and $\mathbf{y}^i$). Consequently, by modeling the joint distribution, the oracle is implicitly learned. During training, the network is provided with $\mathbf{x}_t$ along with various combinations of masks $\mathbf{m}^i$ and their corresponding regional descriptions $\mathbf{y}^i$. In doing so, the network is compelled to establish fine-grained correspondences to accurately predict the noise associated with each mask $\mathbf{m}^i$.

**Modeling of the Data Distribution** $p(\mathbf{x}_t \mid \mathbf{y})$: In our framework, we assume that the various concepts described by the different components of the conditioning variable $\mathbf{y}$ are conditionally independent given[3] $\mathbf{x}_t$ (Liu et al., 2023; Du et al., 2020). Formally, for any distinct indices $i$ and $j$, this assumption implies

$$p(\mathbf{y}^i, \mathbf{y}^j \mid \mathbf{x}_t) = p(\mathbf{y}^i \mid \mathbf{x}_t) \, p(\mathbf{y}^j \mid \mathbf{x}_t). \tag{7}$$

Under this assumption, the joint distribution over $\mathbf{x}_t$ and the set of regional descriptions $\{\mathbf{y}^1, \ldots, \mathbf{y}^n\}$ can be factorized as follows (Liu et al., 2023; Du et al., 2020):

$$p(\mathbf{x}_t \mid \mathbf{y}) \propto p(\mathbf{x}_t, \mathbf{y}^1, \ldots, \mathbf{y}^n) = p(\mathbf{x}_t) \prod_{i=1}^n p(\mathbf{y}^i \mid \mathbf{x}_t). \tag{8}$$

---

[2]Flow matching also shares the same assumption described in Equation 2, and thus cannot achieve fine-grained text-video alignment either.

[3]This assumption is reasonable in our setting—each $\mathbf{y}^i$ refers to a distinct object or concept—but it does not always hold. In practice, we approximate $\text{CMI}(\mathbf{y}^i; \mathbf{y}^j \mid \mathbf{x}_0)$ from model-predicted probabilities and use it to dynamically reweight the loss; see Sec. 3.3 for details.

By Bayesian rules, we have $p(\mathbf{y}^i \mid \mathbf{x}_t) \propto \frac{p(\mathbf{x}_t \mid \mathbf{y}^i)}{p(\mathbf{x}_t)}$, thus:

$$p(\mathbf{x}_t \mid \mathbf{y}) \propto p(\mathbf{x}_t) \prod_{i=1}^{n} p(\mathbf{y}^i \mid \mathbf{x}_t) \propto p(\mathbf{x}_t) \prod_{i=1}^{n} \frac{p(\mathbf{x}_t \mid \mathbf{y}^i)}{p(\mathbf{x}_t)}. \tag{9}$$

Thus, by modeling the conditional distributions $p(\mathbf{x}_t \mid \mathbf{y}^i)$ for each individual $\mathbf{y}^i$, we effectively capture the overall distribution $p(\mathbf{x}_t \mid \mathbf{y})$. As demonstrated in Equation 6, when modeling the joint distribution $p(\mathbf{x}_t, \mathbf{m}_t^i \mid \mathbf{y}^i)$, the conditional distribution $p(\mathbf{x}_t \mid \mathbf{y}^i)$ is concurrently learned. Consequently, the factorization expressed in Equation 8 ensures that the generative capability of the model is maintained by effectively representing $p(\mathbf{x}_t \mid \mathbf{y})$.

**Joint Distribution Modeling**: In order to render $p(\mathbf{x}_t, \mathbf{m}_t^i \mid \mathbf{y}^i)$ tractable, we condition on a known endpoint $(\mathbf{x}_0, \mathbf{m}_0)$ and marginalize over it, as indicated in Equation 1. We compute $p(\mathbf{x}_t, \mathbf{m}_t^i \mid \mathbf{y}^i)$ via:

$$\int p(\mathbf{x}_t, \mathbf{m}_t^i \mid \mathbf{x}_0, \mathbf{m}_0^i, \mathbf{y}^i) \, p(\mathbf{x}_0, \mathbf{m}_0^i \mid \mathbf{y}^i) \, d(\mathbf{x}_0, \mathbf{m}_0^i).$$

Following the assumption in Equation 2, we obtain:

$$p(\mathbf{x}_t, \mathbf{m}_t^i \mid \mathbf{x}_0, \mathbf{m}_0, \mathbf{y}^i) = \mathcal{N}\left(\sqrt{\bar{\alpha}_t}(\mathbf{x}_0, \mathbf{m}_0), (1 - \bar{\alpha}_t)\boldsymbol{I}\right). \tag{10}$$

It is important to note that although the noise injected is still conditionally independent of $\mathbf{y}^i$ given $(\mathbf{x}_0, \mathbf{m}_0^i)$, the fine-grained conditioning information from $\mathbf{y}^i$ is incorporated through the regional mask $\mathbf{m}_0^i$. In other words, while we retain the conditional independence assumption for the sake of tractability, we effectively integrate $\mathbf{y}^i$ by jointly modeling $\mathbf{x}_t$ and its corresponding region $\mathbf{m}_0^i$. Utilizing the $\epsilon$-parameterization, we define our fine-grained loss as follows:

$$\mathcal{L}_{\text{fine}} = \mathbb{E}_{\mathbf{x}_0, (\mathbf{y}^i, \mathbf{m}^i), t, \epsilon}\left[\left\|\epsilon_\theta\left(\mathbf{x}_t, \mathbf{m}_t^i, t, \mathbf{y}^i\right) - \epsilon\right\|_2^2\right], \tag{11}$$

where $\epsilon$ denotes the noise introduced during the forward process. Our joint training not only encourages the model to accurately predict the noise component but also enforces a fine-grained correspondence between the generated video $\mathbf{x}_t$, its associated region $\mathbf{m}^i$, and the conditioning information $\mathbf{y}^i$. By doing so, the model is better equipped to capture detailed relationships between regional descriptions and visual content.

### 3.3 PRACTICAL IMPLEMENTATION

**Training Scheme**. As illustrated in the top of Figure 2, during training, for a given video $\mathbf{x}_0$, we randomly sample a set of masks $\{\mathbf{m}_0^i\}$ along with their corresponding regional captions $\{\mathbf{y}^i\}$. These masks and captions are subsequently combined into a single mask-caption pair. Specifically, the individual masks are concatenated to form a unified mask, while the regional captions are concatenated into a single prompt. Noise is then added to both the mask and the input video (sampling independently), and the resulting noisy pairs are processed by the network. Considering the strong correlation between the mask and the video, rather than increasing the latent dimension, we opt to inflate only the input and output projection layers while keeping the original latent dimension unchanged. This strategy introduces only a minimal number of additional parameters while enabling the model to simultaneously generate both the video and the corresponding mask. To facilitate a gradual adaptation of the network for mask generation, we initialize the inflated input and output projection layers with zeros. The entire network is fully trainable to facilitate joint learning of video and mask generation.

**Dynamic Loss Weighting.** In our framework, to recover the original training objective conditioned on the full prompt $\mathbf{y}$, we invoke a mild conditional independence assumption (Equation. 7) among regional captions. However, this assumption does not always hold in practice, particularly when distinct objects exhibit strong correlations or interactions—for instance, "raining" and "umbrella" frequently co-occur. To address this, we quantify violations of the assumption and adaptively modulate the loss to revert toward the standard diffusion objective when the assumption is weak. Specifically, we assess the conditional independence by estimating the conditional mutual information (CMI) using a fixed, pre-trained CLIP encoder (Radford et al., 2021). We embed the clean video $\mathbf{x}_0$ via its middle frame to derive a global image embedding $\mathbf{e}_{\mathbf{x}_0}$. The textual region descriptions $\mathbf{y}^i$ and $\mathbf{y}^j$ are

Table 1: Quantitative results on VBench, focusing on Multiple Object, Object Class, Color, Scene, Human Action, Spatial Relation, and Overall Consistency.

| Model | Multiple Object | Object Class | Color | Scene | Human Action | Spatial Relation | Overall Consistency | Appearance Style |
|---|---|---|---|---|---|---|---|---|
| VideoCrafter2 | 40.66 | 92.55 | 92.92 | 55.29 | 95.00 | 35.86 | 28.23 | 25.13 |
| HunyuanVideo | 68.55 | 86.10 | 91.60 | 53.88 | 94.40 | 68.68 | 26.44 | 19.80 |
| CogVideoX-5B | 62.11 | 85.23 | 82.81 | 53.20 | 99.40 | 66.35 | 27.59 | 24.91 |
| Sora | 70.85 | 93.93 | 80.11 | 56.95 | 98.20 | 74.29 | 26.26 | 24.76 |
| Gen-3 | 53.64 | 87.81 | 80.90 | 54.57 | 96.40 | 65.09 | 26.69 | 24.31 |
| Kling | 68.05 | 87.24 | 89.90 | 50.86 | 93.40 | 73.03 | 26.42 | 19.62 |
| ModelScope | 38.98 | 82.25 | 81.72 | 39.26 | **92.40** | 33.68 | 25.67 | 23.39 |
| +JDM | **43.02** | **89.35** | **82.16** | **42.91** | 91.20 | **34.70** | **26.11** | **23.71** |
| CogVideoX-2B | 62.63 | 83.37 | 79.41 | 51.14 | 98.00 | 69.90 | 26.66 | **24.80** |
| +JDM | **72.34** | **94.08** | **82.60** | **54.68** | **98.20** | **74.86** | **27.51** | 24.04 |

| Model | Temporal Flickering | Motion Smoothness | Dynamic Degree | Aesthetic Quality | Subject Consistency | Background Consistency | Imaging Quality | Temporal Style |
|---|---|---|---|---|---|---|---|---|
| VideoCrafter2 | 98.41 | 97.73 | 42.50 | 63.13 | 96.85 | 98.22 | 67.22 | 25.84 |
| HunyuanVideo | 99.44 | 98.99 | 70.83 | 60.36 | 97.37 | 97.76 | 67.56 | 23.89 |
| CogVideoX-5B | 98.66 | 96.92 | 70.97 | 61.98 | 96.23 | 96.52 | 62.90 | 25.38 |
| Sora | 98.87 | 98.74 | 79.91 | 63.46 | 96.23 | 96.35 | 68.28 | 25.01 |
| Gen-3 | 98.61 | 99.23 | 60.14 | 63.34 | 97.10 | 96.62 | 66.82 | 24.71 |
| Kling | 99.30 | 99.40 | 46.94 | 61.21 | 98.33 | 97.60 | 65.62 | 24.17 |
| ModelScope | **98.28** | 95.79 | **66.39** | 52.06 | 89.87 | 95.29 | 58.57 | 25.37 |
| +JDM | 98.27 | **98.04** | 63.15 | **54.62** | **96.24** | **98.34** | **60.34** | **25.71** |
| CogVideoX-2B | 98.89 | **97.73** | 59.86 | 60.82 | **96.78** | 96.63 | **61.68** | 24.36 |
| +JDM | **98.99** | 97.36 | **62.08** | **62.69** | 93.12 | **96.82** | 60.32 | **25.64** |

encoded by CLIP's text encoder into $\mathbf{e}_{y^i}$ and $\mathbf{e}_{y^j}$. We then compute the similarity of the $\mathbf{e}_{y^j}$ with $\mathbf{e}_{y^i}$ given condition $\mathbf{e}_{\mathbf{x}_0}$. Conditioning on $\mathbf{x}_0$ is achieved by concatenating $\mathbf{e}_{\mathbf{x}_0}$ to each text embedding prior to computing similarity; this context-augmented cosine similarity is denoted $\text{sim}(\cdot, \cdot \mid \mathbf{e}_{\mathbf{x}_0})$. We approximate $I(\mathbf{y}^i; \mathbf{y}^j \mid \mathbf{x}_0)$ using an InfoNCE-style lower bound, contrasting the joint alignment of $(\mathbf{y}^i, \mathbf{y}^j)$ against a Monte Carlo marginal obtained by shuffling $\mathbf{y}^j$ across similar contexts. For each triplet $(\mathbf{y}^i, \mathbf{y}^j, \mathbf{x}_0)$, we sample $M = 1024$ negatives $\{\mathbf{y}^{j(k)}\}_{k=1}^{M}$ from other videos whose global frame embeddings are nearest neighbors to $\mathbf{e}_{\mathbf{x}_0}$ under CLIP cosine similarity, yielding:

$$\hat{I}(\mathbf{y}^i; \mathbf{y}^j \mid \mathbf{x}_0) \approx \mathbb{E}\left[ \log \frac{\text{sim}(\mathbf{e}_{y^i}, \mathbf{e}_{y^j} \mid \mathbf{e}_{x_0})}{\frac{1}{M}\sum_{k=1}^{M} \text{sim}(\mathbf{e}_{y^i}, \mathbf{e}_{y^{j(k)}} \mid \mathbf{e}_{x_0})} \right]. \tag{12}$$

We precompute the CMI for all videos in the training set. We then dynamically weight the loss using the precomputed CMI to revert toward the standard diffusion loss when the conditional independence assumption is violated. Specifically, we define the dynamic weight as $w' = w \cdot \exp(-\alpha \cdot \bar{I})$, where $w \in [0, 1]$ is a base hyperparameter (set to 1.0 in our experiments), $\alpha > 0$ controls sensitivity (set to 1.0 in our experiments), and $\bar{I}$ is the average CMI across pairs in the sample. The joint loss is then given by

$$\mathcal{L} = (1 - w') \mathcal{L}_{\text{diff}} + w' \mathcal{L}_{\text{fine}}. \tag{13}$$

When $\mathbf{y}^i$ and $\mathbf{y}^j$ are highly related given $\mathbf{x}_0$, $\hat{I}(\mathbf{y}^i; \mathbf{y}^j \mid \mathbf{x}_0)$ becomes large, causing $w'$ to approach zero and the loss to fall back to the standard diffusion objective. Conversely, if they are weakly related given $\mathbf{x}_0$, $\hat{I}(\mathbf{y}^i; \mathbf{y}^j \mid \mathbf{x}_0)$ approaches zero, yielding $w' \approx 1$ and emphasizing the fine-grained loss. This formulation prioritizes $\mathcal{L}_{\text{fine}}$ for low-CMI samples (weak dependencies) while downweighting it for high-dependence cases, ensuring robustness.

## 4 EXPERIMENTS

To evaluate the performance of our proposed method, we fine-tuned two open-sourced text-to-video generation models: CogVideoX-2B, based on the DiT architecture, and ModelScopeT2V, based on a UNet architecture and compare the methods with existing baselines (Details in Appendix. A.7).

**Zero-shot Text-to-Video Generation on VBench.** VBench (Huang et al., 2023b) evaluates text-video alignment across 16 dimensions on approximately 5,000 videos. We focus on fine-grained

**CogvideoX-2B**          **CogvideoX-2B + JDM**          **Generated Mask**          **Detected Mask**

A vibrant orange carrot with lush green leaves stands upright on a wooden table, bathed in soft, natural light. Beside it, a colorful umbrella with a whimsical pattern of raindrops ...

A vibrant orange sits on a rustic wooden table, its bright color contrasting with the aged wood. Beside it, an antique clock with a brass frame and Roman numerals ticks softly, its hands moving steadily, ...

A sleek, black motorcycle with chrome accents speeds down a bustling city street, its rider wearing a leather jacket and helmet, reflecting the urban lights. In the background, a vibrant yellow bus, ...

A sleek, modern smartphone with a glossy black finish lies on a rustic wooden table, its screen reflecting ambient light. Beside it, a vibrant red apple with a perfect sheen sits, contrasting the technology...

A rustic wooden table holds a ceramic bowl filled with vibrant, fresh fruit, including apples, oranges, and grapes, their colors popping against the natural wood grain. Beside the bowl, a sleek, modern remote control rests, its black surface contrasting with the organic textures around it, ...

Figure 3: Qualitative results. We compare the baseline CogVideoX-2B with our proposed method (CogVideoX-2B + JDM) (*Click to play, best viewed with Acrobat Reader*).

metrics: Multiple Object, Object Class, Color, Scene, Human Action, Spatial Relation, and Overall Consistency. Table 1 shows that JDM leads to substantial improvements. For ModelScopeT2V, scores increase in Multiple Object, Object Class, Color, Scene, and Overall Consistency, with slight decreases in Human Action. For CogVideoX-2B+JDM, gains are more pronounced across all these metrics. In secondary metrics, JDM generally enhances quality (e.g., Aesthetic Quality), with occasional minor trade-offs like reduced Subject Consistency. These results confirm JDM's effectiveness in improving fine-grained alignment while maintaining visual coherence.

**Zero-Shot Text-to-Video Generation on T2VCompBench.** T2VCompBench (Sun et al., 2025) is designed to evaluate the compositional capabilities of text-to-video generation using 1,400 diverse prompts. The benchmark focuses on challenging scenarios where correct binding of attributes and actions is crucial. Metrics such as *Consistent Attribute Binding*, *Action Binding*, and *Motion Binding* assess whether objects and their corresponding attributes or actions are generated and associated correctly. As shown in Table 2, JDM significantly improves performance. For instance, Mod-

elScopeT2V improves in *Consistent Attribute Binding* from 0.5148 to 0.5684, in *Motion Binding* from 0.2408 to 0.2468, and in *Action Binding* from 0.3639 to 0.4016; CogVideoX-2B improves in *Consistent Attribute Binding* from 0.6174 to 0.7067, in *Motion Binding* from 0.2612 to 0.2735, and in *Action Binding* from 0.5039 to 0.5690. The results demonstrate that JDM significantly improve the fine-grained text-video alignment.

**Mask Quality Evaluation.** To evaluate the quality of our generated masks, we randomly selected 100 masks generated from the VBench prompts. For each video, a human annotator manually delineated the object mask corresponding to the text prompt on the first frame. Subsequently, Grounded-SAM2 was employed to detect and propagate the mask across the entire video. Table 3 presents quantitative results on mask quality for two models that are capable of generating both video and mask. Specifically, the ModelScopeT2V+JDM model achieves an IoU of

Table 2: Quantitative Results on T2VCompBench

| Model | Consist Attribute Binding | Motion Binding | Action Binding |
|---|---|---|---|
| VideoCrafter2 | 0.6182 | 0.2259 | 0.5030 |
| CogVideoX-5B | 0.6164 | 0.2658 | 0.5333 |
| Mochi | 0.5973 | 0.2334 | 0.4759 |
| Gen-3 | 0.5980 | 0.2754 | 0.5233 |
| ModelScopeT2V | 0.5148 | 0.2408 | 0.3639 |
| +JDM | **0.5684** | **0.2468** | **0.4016** |
| CogVideoX-2B | 0.6174 | 0.2612 | 0.5039 |
| +JDM | **0.7067** | **0.2735** | **0.5690** |

0.3264, an F1 score of 0.3348, and a Pixel Accuracy of 0.4262. These relatively low values suggest that the mask quality generated by this model is suboptimal, likely due to its limited capacity and older architecture. In contrast, the CogVideoX-2B+JDM model attains much higher performance with an IoU of 0.7141, an F1 score of 0.7561, and a Pixel Accuracy of 0.8031, indicating a stronger ability to capture fine-grained details in the mask.

**Qualitative Result.** As shown in Figure 3, we present qualitative results on prompts describing multiple objects and compare our method with the CogVideoX-2B baseline. We also include the masks generated from our videos alongside the masks detected by applying Grounded-SAM2 to our generated content. It is evident that by employing JDM, our model is significantly more effective at gener-

Table 3: Quantitative Results on Mask Quality

| Metric | ModelScopeT2V +JDM | CogVideoX-2B +JDM |
|---|---|---|
| IoU | 0.3264 | 0.7141 |
| F1 | 0.3348 | 0.7561 |
| Pixel Acc | 0.4262 | 0.8031 |

ating multiple objects simultaneously. In contrast, the baseline CogVideoX-2B tends to generate only one of the described objects or produces objects that are truncated at the edges of the video. For instance, in Figure 3(e), the baseline fails to generate the remote entirely, while in Figure 3(b), it generates the orange without the clock. Similarly, in Figures 3(c) and (d), some objects are truncated at the boundaries, underscoring the baseline's limitations in fine-grained text-video alignment. Overall, these qualitative results demonstrate that the incorporation of JDM significantly enhances compositional text-to-video generation and improves fine-grained text-video alignment.

**User Study.** To further validate our approach, we conducted a user study in which 20 participants evaluated 15 video pairs generated by the baseline and the JDM-enhanced models. For each pair, participants selected one of three options: (A) baseline video, (B) JDM video, or (C) "cannot decide." Table 4 reports the percentages of votes for the baseline and JDM-enhanced versions (the remaining votes indicate indecision). Notably, for text-video alignment, 48.5% and 51.7% of the votes favored the JDM-enhanced videos for ModelScopeT2V and CogVideoX-2B, respectively. Similarly, for visual quality, 58.0% and 45.5% of votes were cast for the JDM-enhanced versions. These findings confirm that our JDM approach significantly improves text-video alignment without the loss of visual quality.

**Ablation Study.** To address potential biases from our dataset filtering, we performed an ablation study comparing our Joint Diffusion Model (JDM) against direct fine-tuning on the same filtered dataset. Both approaches were evaluated on VBench, emphasizing metrics for multiple ob-

Table 4: User Study Preferences (%)

| Metric | Baseline | JDM | Indecision |
|---|---|---|---|
| Semantic Alignment (ModelScopeT2V) | 10.20 | **48.50** | 41.30 |
| Semantic Alignment (CogVideoX-2B) | 20.17 | **51.68** | 28.15 |
| Visual Quality (ModelScopeT2V) | 21.86 | **58.04** | 20.10 |
| Visual Quality (CogVideoX-2B) | 34.37 | **45.50** | 20.13 |

jects, object class, color, scene, spatial relation, and overall consistency. Performance was assessed

every 500 training steps, as shown in Figure 4. Direct fine-tuning yielded no notable gains in fine-grained text-video alignment, with metrics fluctuating around baseline levels. In contrast, JDM consistently improved these metrics throughout training, underscoring the efficacy of our method.

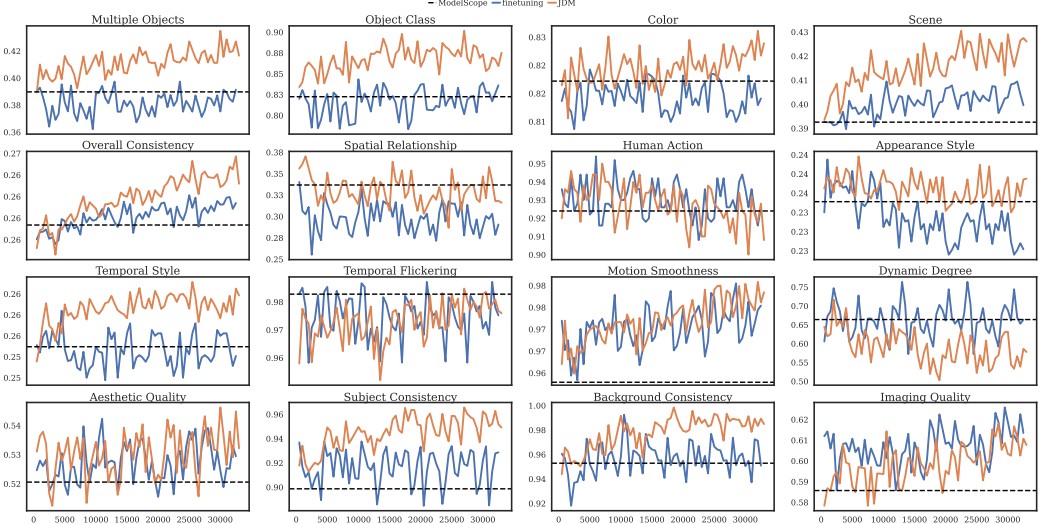

Figure 4: Ablation study. We fine-tune the base model on the same dataset and perform zero-shot evaluation on VBench. Metrics are recorded every 500 steps, and the curves compare performance over training.

## 5 CONCLUSION

In this paper, we introduce the JDM framework for fine-grained text-to-video generation. We reveal that conventional video diffusion models often struggle to accurately capture detailed textual instructions, primarily due to a training objective that emphasizes video reconstruction over explicit text-video alignment. By modeling the joint distribution of video content and its corresponding mask, JDM directly enforces fine-grained alignment between visual elements and input text prompts. Our experimental results, obtained by integrating JDM into two distinct text-to-video models, demonstrate substantial improvements in text-video alignment while preserving high video quality. Furthermore, the concurrent generation of video and mask unlocks new avenues for tasks such as simultaneous synthesis and segmentation. Looking ahead, future work could extend JDM to handle more complex multi-object interactions or incorporate real-time inference for interactive applications, further enhancing its utility in creative and practical domains.

**Ethics statement** This work adheres to the ICLR Code of Ethics. Our research does not involve human subjects, studies with potential for harm, or methodologies raising concerns regarding discrimination, bias, fairness, privacy, or security. No human-annotated datasets were used in the process; all data processing and model training rely on publicly available or synthetically generated resources in compliance with legal and ethical standards. We have ensured research integrity through rigorous documentation and reproducibility efforts, as detailed in the Reproducibility Statement.

**Reproducibility Statement** To facilitate reproducibility of our results, we provide comprehensive details on the training parameters in Appendix A.7, the network architecture in Appendix 3.1, the joint modeling approach in Section. 3.2, and the theoretical derivations in Appendix A.5. Furthermore, we outline the dataset construction process in Appendix A.6.

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

# A APPENDIX

## A.1 LLM USAGE

In preparing this manuscript, we utilized ChatGPT and Grok, solely for language polishing and minor refinements to improve clarity, grammar, and flow in the text. The LLM was provided with sections of the draft and asked to suggest revisions, which were then reviewed, edited, and incorporated by the authors as deemed appropriate. All core ideas, research contributions, technical details, and analyses originate from the authors and were not generated or ideated by the LLM. No other LLMs were used in the research process.

## A.2 CONSTRUCTING FINE-GRAINED TEXT-VIDEO CORRESPONDENCE

As illustrated in the bottom panel of Figure 2, our method for establishing fine-grained correspondence between text and video proceeds through several systematic stages. First, given an input video, we employ CogVLM2 (Hong et al., 2024) to detect and identify salient objects across frames. Next, for each detected object, Grounded-SAM2 (Ren et al., 2024) is utilized to extract and track a precise segmentation mask over time. To ensure reliable text-video alignment, we apply a strict mask quality filtering process. Specifically, we filter out: (1) masks truncated by frame boundaries, (2) masks that are too large (¿0.6 of frame area) or too small (¡0.1 of frame area), (3) videos with more than 10 objects, as these typically result in ambiguous correspondence, and (4) masks that are disconnected or fragmented across frames. Finally, for every remaining mask $\mathbf{m}^i$, we prompt BLIP2 (Li et al., 2023a) to generate a corresponding regional caption $\mathbf{y}^i$, thereby associating detailed textual descriptions with specific visual regions.

## A.3 DISCUSSION ON ALTERNATIVE REGIONAL SIGNALS OTHER THAN MASKS

Our primary method for enabling fine-grained text-to-video generation is the incorporation of text-regional correspondence during the generation process. We selected object masks for their simplicity and sufficiency in encoding spatial correspondence. While other spatial signals such as depth and HED can also provide effective regional signals, they include additional low-level details that may be unnecessary for learning text-region alignment. As shown in Table 5, we applied JDM to CogVideoX-2B using **regional** HED and depth as auxiliary supervision, where these signals correspond to regional text descriptions (consistent with our framework). Specifically, regional HED and depth signals are obtained by first extracting whole-scene HED and depth maps, then cropping them according to the regional masks. We trained these three variants for the same number of optimizer steps and evaluated them on VBench. As illustrated in the table, the JDM-HED and JDM-Depth variants offer similar advantages to the mask version, albeit with slightly lower performance, confirming that masks provide a more minimal and sufficient representation for regional correspondence.

| Variant | Mul Obj | Obj Class | Color | Scene | Human Action | Consistency | Spatial Rel |
|---|---|---|---|---|---|---|---|
| JDM-Mask | 72.34+15.50% | 94.08+12.85% | 82.60+4.02% | 54.68+6.92% | 98.20+0.20% | 27.97+4.91% | 74.86+7.10% |
| JDM-HED | 70.32+12.28% | 92.86+11.38% | 81.60+2.76% | 53.71+5.03% | 98.40+0.41% | 27.02+1.35% | 73.08+4.55% |
| JDM-Depth | 72.35+15.52% | 93.28+11.89% | 82.52+3.92% | 54.21+6.00% | 98.60+0.61% | 27.32+2.48% | 72.24+3.35% |

Table 5: VBench results with CogVideoX-2B-JDM variants

## A.4 COMPUTATIONAL EFFICIENCY OF JDM

Our JDM implementation inflates the input/output projection layers to jointly produce video and mask outputs. To assess the computational cost, we report wall-clock forward time and parameter count on a single NVIDIA A100 (batch size $= 1$). The CogVideoX uses 49 frames at $480 \times 720$ and ModelScope T2V uses 16 frames at $256 \times 256$. As shown in Table 6, JDM introduces negligible parameter overhead ($\leq 0.02\%$) and small runtime overhead ($\leq 2.14\%$), while enabling simultaneous video+mask generation in a single pass.

## A.5 DERIVATION OF DIFFUSION

We start with a navie loss which regress on the conditional score function directly.

$$\mathcal{L}_{\text{naive}} = \mathbb{E}_{t,(\mathbf{x}_t,\mathbf{y})\sim p(\mathbf{x}_t,\mathbf{y})}\Big[\lambda(t)\big\|s_\theta(\mathbf{x}_t,t,\mathbf{y}) - \nabla_{\mathbf{x}_t}\log p_t(\mathbf{x}_t \mid \mathbf{y})\big\|_2^2\Big]. \tag{14}$$

To make $p(\mathbf{x}_t,\mathbf{y})$ tractable, we condition it on $\mathbf{x}_0$ and then marginlize over it,

$$p_t(\mathbf{x}_t \mid \mathbf{y}) = \int p_t(\mathbf{x}_t \mid \mathbf{x}_0,\mathbf{y})\, p(\mathbf{x}_0 \mid \mathbf{y})\, d\mathbf{x}_0. \tag{15}$$

Subsitute the Equation 15 into the score function:

$$\nabla_{\mathbf{x}_t}\log p_t(\mathbf{x}_t \mid \mathbf{y}) = \nabla_{\mathbf{x}_t}\log \int p_t(\mathbf{x}_t \mid \mathbf{x}_0,\mathbf{y})\, p(\mathbf{x}_0 \mid \mathbf{y})\, d\mathbf{x}_0,$$

we differentiate under the integral (assuming appropriate regularity conditions) to obtain:

$$\begin{aligned}
\nabla_{\mathbf{x}_t}\log p_t(\mathbf{x}_t \mid \mathbf{y}) &= \frac{\nabla_{\mathbf{x}_t}\int p_t(\mathbf{x}_t \mid \mathbf{x}_0,\mathbf{y})\, p(\mathbf{x}_0 \mid \mathbf{y})\, d\mathbf{x}_0}{\int p_t(\mathbf{x}_t \mid \mathbf{x}_0,\mathbf{y})\, p(\mathbf{x}_0 \mid \mathbf{y})\, d\mathbf{x}_0} \\
&= \frac{\int p(\mathbf{x}_0 \mid \mathbf{y})\, \nabla_{\mathbf{x}_t} p_t(\mathbf{x}_t \mid \mathbf{x}_0,\mathbf{y})\, d\mathbf{x}_0}{\int p(\mathbf{x}_0 \mid \mathbf{y})\, p_t(\mathbf{x}_t \mid \mathbf{x}_0,\mathbf{y})\, d\mathbf{x}_0} \\
&= \mathbb{E}_{\mathbf{x}_0\sim p(\mathbf{x}_0\mid\mathbf{y},\mathbf{x}_t)}\Big[\nabla_{\mathbf{x}_t}\log p_t(\mathbf{x}_t \mid \mathbf{x}_0,\mathbf{y})\Big].
\end{aligned}$$

Substitue it into Equation 14:

$$\mathcal{L}_{\text{naive}} = \mathbb{E}_{t,(\mathbf{x}_t,\mathbf{y})\sim p(\mathbf{x}_t,\mathbf{y}),\mathbf{x}_0\sim p(\mathbf{x}_0\mid\mathbf{y},\mathbf{x}_t)}\Big[\lambda(t)\big\|s_\theta(\mathbf{x}_t,t,\mathbf{y}) - \nabla_{\mathbf{x}_t}\log p_t(\mathbf{x}_t \mid \mathbf{x}_0,\mathbf{y})\big\|_2^2\Big]. \tag{16}$$

$$= \mathbb{E}_{t,(\mathbf{x}_0,\mathbf{y},\mathbf{x}_t)\sim p(\mathbf{x}_0,\mathbf{y},\mathbf{x}_t)}\Big[\lambda(t)\big\|s_\theta(\mathbf{x}_t,t,\mathbf{y}) - \nabla_{\mathbf{x}_t}\log p_t(\mathbf{x}_t \mid \mathbf{x}_0,\mathbf{y})\big\|_2^2\Big]. \tag{17}$$

$$= \mathbb{E}_{t,(\mathbf{x}_0,\mathbf{y})\sim p(\mathbf{x}_0,\mathbf{y}),\mathbf{x}_t\sim p(\mathbf{x}_t\mid\mathbf{y},\mathbf{x}_0)}\Big[\lambda(t)\big\|s_\theta(\mathbf{x}_t,t,\mathbf{y}) - \nabla_{\mathbf{x}_t}\log p_t(\mathbf{x}_t \mid \mathbf{x}_0,\mathbf{y})\big\|_2^2\Big]. \tag{18}$$

Assuming the perturbation kernel is independent of $\mathbf{y}$ given $\mathbf{x}_0$:

$$p_t(\mathbf{x}_t \mid \mathbf{x}_0,\mathbf{y}) = p_t(\mathbf{x}_t \mid \mathbf{x}_0) = \mathcal{N}(\sqrt{\bar{\alpha}_t}\mathbf{x}_0, \sqrt{1-\bar{\alpha}_t}I). \tag{19}$$

Thus we have:

$$\mathcal{L}_{\text{naive}} = \mathbb{E}_{t,(\mathbf{x}_0,\mathbf{y})\sim p(\mathbf{x}_0,\mathbf{y}),\mathbf{x}_t\sim p(\mathbf{x}_t\mid\mathbf{x}_0)}\Big[\lambda(t)\big\|s_\theta(\mathbf{x}_t,t,\mathbf{y}) - \nabla_{\mathbf{x}_t}\log p_t(\mathbf{x}_t \mid \mathbf{x}_0)\big\|_2^2\Big]. \tag{20}$$

## A.6 DATA PREPROCESSING AND ANNOTATION

We use WebVid-10M as the foundation for our fine-tuning dataset. WebVid-10M contains a highly diverse collection of videos, making it suitable for improving general fine-grained text-video alignment. However, the videos in WebVid-10M often contain watermarks and exhibit relatively low visual quality. Additionally, the captions provided in the dataset are of suboptimal quality. To address these issues, we first recaption the videos using CogVLM2, following the approach of CogVideoX. Specifically, we employ the following prompt for video captioning:

---
**Prompt for Video Captioning**

**Video Captioning Prompt:** *Video Captioning: Please provide a detailed description of this video, focusing on the objects and concepts present. The description should be between 20 and 100 words. The answer is:*

---

Table 6: Overhead of JDM relative to original baselines. Entries are *original → JDM (relative change)*.

| Metric | CogVideoX | ModelScope |
|---|---|---|
| Params (B) | 1.6938 → 1.6940 (+0.015%) | 1.4112 → 1.4113 (+0.002%) |
| Forward time (s) | 0.4946 → 0.5052 (+2.14%) | 0.4226 → 0.4263 (+0.88%) |

Next, to identify objects within the videos, we utilize CogVLM2, a state-of-the-art visual language model, for object recognition. The following prompt is employed to guide the model in extracting object information:

> **Prompt for Object Recognition**
>
> **Object Recognition Prompt:** *What objects are present in this video? List them concisely, separating each object with a comma. Provide only the names of the objects without additional descriptions, numerical values, or temporal details. The output should be:*

This approach ensures a clear and structured extraction of object information, facilitating further analysis and alignment with textual descriptions. We then feed the object names and corresponding videos to Grounded-SAM2 to obtain segmentation masks for each object.

To ensure high-quality fine-grained text-video correspondence—i.e., determining which part of the text corresponds to which part of the video—we apply several filtering steps. First, we filter out videos containing more than two objects of the same type, as this can lead to ambiguous text-video alignment (e.g., the term "a man" might refer to different individuals in the video). Second, we filter out objects with masks that are too small, setting a threshold of 0.1 (i.e., each object must occupy at least 10% of the video region). Third, to simplify the learning process, we filter out videos with an excessive number of objects, retaining only those with between 2 and 10 distinct objects.

After applying these filters, we use the segmentation masks to extract regions of interest from the videos and feed them to BLIP-2 for regional captioning. However, we observe that BLIP-2 tends to describe the black background alongside the regions of interest. To address this, we manually remove all descriptions related to the black background. Following this comprehensive filtering process, we obtain a refined subset of approximately 1 million videos.

### A.7 TRAINING DETAILS AND HYPERPARAMETERS

For CogVideoX-2B, we modify the original input and output channels from 16 to 32 to concatenate the mask along the channel dimension. The model is trained with a learning rate of $5 \times 10^{-5}$, a batch size of 768, and the AdamW optimizer for 10,000 steps. Training is performed on 64 Nvidia A100 GPUs using mixed precision (fp16). During inference, the model generates videos at a resolution of $480 \times 720 \times 49$ with 50 sampling steps.

For ModelScopeT2V, we modify the original input and output channels from 4 to 8 to concatenate the mask along the channel dimension. We train the model using a learning rate of $1 \times 10^{-5}$, a batch size of 960, and the AdamW optimizer for 30,000 steps. The model is trained at a resolution of $256 \times 256 \times 16$, and we employ DeepSpeed Zero Stage 2 with CPU offloading to optimize memory usage during training. Training is conducted on a cluster of 80 Nvidia RTX 4090 GPUs. We utilize the OneCycle scheduler for learning rate scheduling and train the model using Bfloat16 precision. Additionally, we apply a 10% dropout rate for text conditioning to enhance generalization. During inference, we use the DDIM sampler with 50 steps and a classifier-free guidance scale of 9, following the original paper.

## A.8 QUALITATIVE COMPARISON WITH MODELSCOPET2V

ModelScopeT2V        ModelScopeT2V+JDM        ModelScopeT2V        ModelScopeT2V+JDM

(a) A backpack and an umbrella.                    (b) A bear and a zebra.

(c) A bicycle and a car.                    (d) A book and a clock.

(e) A bottle and a chair.                    (f) A bowl and a remote.

(g) A cake and a vase.                    (h) A car and a motorcycle.

Figure 5: Qualitative comparision between ModelScopeT2V and ModelScopeT2V+JDM

