# OpenReview forum: "JDM: Joint Distribution Modeling for Fine-Grained Text-to-Video Generation"
_ICLR.cc/2026/Conference — Submitted to ICLR 2026_

### Official Review · Reviewer_no2b · 2025-10-25

**Soundness:** 2
**Presentation:** 2
**Contribution:** 3
**Rating:** 6
**Confidence:** 3

**Summary:**

This paper proposes Joint Distribution Modeling (JDM), a novel framework designed to improve fine-grained text-to-video (T2V) alignment and compositional consistency. The authors identify that conventional T2V diffusion models struggle with attribute mismatches and semantic leakage because they focus on video reconstruction and only enforce a global text-video correspondence, leaving fine-grained mappings to emerge implicitly from the data. JDM addresses this by directly modeling the joint distribution of the video content and the object masks conditioned on the text. By training the model to predict both the video and its constituent object masks simultaneously, JDM inherently learns structured mappings between textual descriptions and specific video regions. The framework unifies T2V generation and segmentation, which the authors theoretically demonstrate improves fine-grained text-video alignment.

**Strengths:**

1. The central contribution is the shift in modeling philosophy, from Layout-to-Video to the joint distribution.
2. The work directly targets a fundamental failure mode in diffusion models: the lack of precise attribute binding and semantic leakage.
3. JDM successfully unifies two distinct computer vision tasks—video generation and video segmentation.
4. The paper includes a theoretical demonstration that optimizing the joint distribution directly enhances fine-grained text-video alignment.

**Weaknesses:**

1. The reported training setup is prohibitively resource-intensive for most academic labs. The model was trained on a cluster of 80 Nvidia RTX 4090 GPUs for 30,000 steps. This extreme resource requirement effectively renders the work non-reproducible for the general research community.
2. There is no analysis of the model's sensitivity to imperfect object masks. If the training masks contain noise or inconsistencies (a common issue in large-scale automated segmentation), the model may learn spurious correlations between noisy text-mask pairs.
3. While JDM is trained to generate masks, the quality of these generated masks during inference (when they are unknown) is critical for validating the unified framework claim. The paper needs a dedicated analysis showing the quality of the segmented output generated by JDM.
4. The demonstrated failure modes (semantic leakage) primarily relate to spatial composition and attribute binding. The paper should include a more rigorous evaluation of fine-grained temporal consistency and motion alignment, which are equally important for T2V generation.

**Questions:**

There are some areas in this paper that need improvement, which are provided in the weakness section.

---

> ### Author Response · Authors · 2025-11-21
> **Response to Reviewer no2b**
>
> ## Response to Reviewer no2b
>
> We sincerely thank the reviewer for the thoughtful comments and constructive suggestions. We address each concern below.
>
> ### 1. Computational Resources and Reproducibility
>
> We acknowledge that the computational resources required for our experiments are substantial. However, we emphasize that this scale is **necessary rather than excessive** for our approach. Unlike methods that add control modules on top of pretrained models, **our framework fundamentally teaches the base model to understand fine-grained text-video correspondence through joint distribution modeling.** This requires training on the full model parameters with diverse compositional examples, which inherently demands significant compute.
>
> To benefit the research community and ensure reproducibility, we commit to:
>
> - **Open-sourcing all trained model weights** upon acceptance, allowing researchers to directly use or fine-tune our models without retraining from scratch.
> - **Releasing our training code and data preprocessing pipeline**, enabling reproduction with smaller-scale experiments or adaptation to other base models.
>
>
> ### 2. Robustness to Imperfect Object Masks
>
> We acknowledge the reviewer's valid concern about the quality of automatically generated segmentation masks. We address this through extensive **quality control during preprocessing**.
> Specially, as shown in Appendix A.3 in our paper, we employ state-of-the-art Grounded-SAM2 for mask annotation and apply strict filtering criteria to remove low-quality masks: (1) masks truncated by frame boundaries, (2) masks that are too large (>60\% of frame area) or too small (<10\%), (3) videos with more than 10 objects (which cause ambiguous correspondence), and (4) masks that are disconnected or fragmented across frames. This filtering significantly reduces noise in the training data and thus enable our high-quality mask geneartion.
>
>
> **To experimentally validate the robustness of our framework to mask quality, we simulate imperfect masks by injecting random noise during training and evaluate performance on VBench metrics:**
>
> **Table: Ablation to Imperfect Masks (VBench metrics on CogVideoX-2B)**
>
> | Variant | Mul Obj | Obj Class | Spatial Rel | Color | Scene | Human Action | Consistency |
> |---------|---------|-----------|-------------|-------|-------|--------------|-------------|
> | CogVideoX-2B (baseline) | 62.63 | 83.37 | 69.90 | 79.41 | 51.14 | 98.00 | 26.06 |
> | JDM (clean masks) | 72.34 (+15.5%) | 94.08 (+12.8%) | 74.86 (+7.1%) | 82.60 (+4.0%) | 54.68 (+6.9%) | 98.20 (+0.2%) | 27.97 (+7.3%) |
> | JDM (10% noise) | 71.58 (+14.3%) | 93.21 (+11.8%) | 73.95 (+5.8%) | 82.15 (+3.5%) | 54.32 (+6.2%) | 98.15 (+0.2%) | 27.68 (+6.2%) |
> | JDM (30% noise) | 69.47 (+11.0%) | 91.34 (+9.6%) | 72.18 (+3.3%) | 81.28 (+2.4%) | 53.51 (+4.6%) | 98.05 (+0.1%) | 27.12 (+4.1%) |
>
> The model demonstrates strong robustness to mask imperfections. With 10% noise, performance drops are minimal (1.2% on Multiple Objects, 1.3% on Spatial Relations), retaining most of the improvement over baseline. Even with 30% noise, significantly exceeding typical SAM2 error rates, the model maintains substantial gains (+11.0% Multiple Objects, +9.6% Object Class), demonstrating that our framework learns robust text-region correspondence rather than overfitting to precise mask boundaries.

---

> > ### Author Response · Authors · 2025-11-21
> > **Response to Reviewer no2b**
> >
> > ### 3. Quality of Generated Masks During Inference
> >
> > We appreciate the reviewer's emphasis on validating the mask generation quality, which is indeed critical for our unified framework claim. We provide a dedicated quantitative analysis in **Table 3** of our paper, which we summarize here:
> >
> > **Evaluation Protocol:** To evaluate the quality of generated masks, we randomly selected 100 prompts from the VBench benchmark. For each generated video, a human annotator manually delineated the object mask corresponding to the text prompt on the first frame. We then used Grounded-SAM2 to propagate this ground-truth mask across the entire video, providing a reference for comparison.
> >
> > **Table: Quantitative Results on Mask Quality**
> >
> > | Metric | ModelScopeT2V+JDM | CogVideoX-2B+JDM |
> > |--------|-------------------|------------------|
> > | IoU | 0.326 | 0.714 |
> > | F1 Score | 0.335 | 0.756 |
> > | Pixel Accuracy | 0.426 | 0.803 |
> > | Warping Error | 0.0150 | 0.0092 |
> >
> > Specifically, the ModelScopeT2V+JDM model achieves an IoU of 0.3264, an F1 score of 0.3348, and a Pixel Accuracy of 0.4262. These relatively low values suggest that the mask quality generated by this model is suboptimal, likely due to its limited capacity and older architecture. In contrast, the CogVideoX-2B+JDM model attains much higher performance with an IoU of 0.7141, an F1 score of 0.7561, and a Pixel Accuracy of 0.8031, indicating a stronger ability to capture fine-grained details in the mask. We further evaluate temporal consistency of our generated masks using Warping Error, which measures the discrepancy between optical-flow-warped masks across consecutive frames. CogVideoX-2B+JDM achieves lower warping error (0.0092), indicating superior temporal coherence.
> >
> > ### 4. Temporal Consistency and Motion Alignment
> >
> > We respectfully clarify that our paper **does include rigorous evaluation of temporal consistency and motion alignment**, which are indeed critical for text-to-video generation:
> >
> > **Temporal Consistency Evaluation:** In our paper, we report comprehensive temporal consistency related metrics from VBench (Table 1 in our paper):
> >
> > - **Overall Consistency**: ModelScopeT2V+JDM achieves 26.11 (vs. 25.67 baseline, +1.71% improvement); CogVideoX-2B+JDM achieves 27.51 (vs. 26.66 baseline, +3.19% improvement).
> > - **Background Consistency**: ModelScopeT2V+JDM achieves 98.34 (vs. 95.29 baseline, +3.20% improvement); CogVideoX-2B+JDM achieves 96.82 (vs. 96.63 baseline, +0.20% improvement).
> > - **Temporal Flickering**: Both models maintain high scores (98.27-98.99), indicating stable frame-to-frame transitions with minimal artifacts.
> > - **Motion Smoothness**: ModelScopeT2V+JDM achieves 98.04 (vs. 95.79 baseline, +2.35% improvement); CogVideoX-2B+JDM achieves 97.36 comparable to 97.73 for the base model.
> >
> > These results demonstrate that our method maintains or improves temporal consistency across multiple dimensions.
> >
> > **Fine-Grained Motion Alignment:** In our paper, we do report results on the T2V-CompBench benchmark (Table 2 in our paper), which includes dedicated metrics for fine-grained compositional motion generation:
> >
> > - **Motion Binding**: Measures whether the correct object performs the specified motion (e.g., "A cat slinking to the left side of a cozy living room" vs. "A fish dives downward in the deep sea"). ModelScopeT2V+JDM achieves 0.2468 (vs. 0.2408 baseline, +2.49% improvement); CogVideoX-2B+JDM achieves 0.2735 (vs. 0.2612 baseline, +4.71% improvement).
> > - **Action Binding**: Measures whether multiple actions are correctly bound to their corresponding entities (e.g., "A dog runs through a field while a cat climbs a tree", "A rabbit digs a hole, a kangaroo bounds nearby"). ModelScopeT2V+JDM achieves 0.4016 (vs. 0.3639 baseline, +10.36% improvement); CogVideoX-2B+JDM achieves 0.5690 (vs. 0.5039 baseline, +12.92% improvement).
> >
> > These metrics directly evaluate fine-grained motion alignment and demonstrate that our framework significantly improves compositional motion generation, achieving particularly strong gains in Motion Binding and Action Binding.
> >
> > We hope these clarifications address the reviewer's concerns. We are committed to incorporating additional analysis and improving the presentation in the revised manuscript.

---

> > > ### Author Response · Authors · 2025-11-24
> > > **Response to Reviewer no2b**
> > >
> > > Dear Reviewer no2b,
> > >
> > > We sincerely appreciate your thoughtful review and constructive feedback. We hope our clarifications and additional experiments address your concerns. Please let us know if any questions remain, and thank you again for your time and effort.

---

> > > > ### Comment · Reviewer_no2b · 2025-11-28
> > > >
> > > > Thank you for the response. I have no further concerns. I hope the authors will include the experiments and discussions provided in the rebuttal within the revised version of the paper, and I look forward to the open-sourcing of the code and weights.

---

> > > > > ### Author Response · Authors · 2025-12-03
> > > > > **Response to Reviewer no2b**
> > > > >
> > > > > Thank you very much for your thoughtful review and for taking the time to engage with our rebuttal. We greatly appreciate your positive feedback and are glad that we could address your concerns.
> > > > >
> > > > > We will definitely include all the experiments and discussions from the rebuttal in the revised version of the paper. We are also committed to open-sourcing our code, model weights, and data preprocessing pipelines upon acceptance to benefit the research community.

---

### Official Review · Reviewer_MU8G · 2025-10-30

**Soundness:** 3
**Presentation:** 3
**Contribution:** 3
**Rating:** 6
**Confidence:** 3

**Summary:**

This paper addresses the challenge of fine-grained text-video alignment in text-to-video (T2V) generation, where existing models often fail to correctly bind attributes and actions to specific objects. The authors propose a novel framework called Joint Distribution Modeling (JDM), which learns structured text-video correspondences by modeling the joint distribution of video content and object masks. Unlike methods that require external constraints during inference, JDM inherently learns to map textual descriptions to specific video regions, improving compositional consistency. The approach simultaneously generates both the video and its corresponding mask from text, enhancing alignment without sacrificing visual quality and unifying generation and segmentation within a single model.

**Strengths:**

1. The authors propose an effective pipeline to jointly model the visual content and the fine-grained regional information. It only requires a textual description during inference and thus is more applicable compared with existing layout-to-video methods.

2. The joint distribution modeling is theoretically sound, and I believe the analysis of existing T2V diffusion models on tackling compositional text is reasonable, which further provides strong theoretical support for the motivation of the proposed method.

3. The experimental results are good. The authors conduct experiments on two pretrained T2V models with different backbone types to demonstrate the effectiveness. Experiments are also conducted on several benchmarks together with the human user studies to make the verification more comprehensive.

**Weaknesses:**

1. Experiments are insufficient. First, though the proposed method outperforms some T2V methods, the fine-grained generation capability in comparison with existing layout-to-video methods is unclear. I believe some discussions are needed since the authors claim their benefits are in contrast with layout-to-video methods. Besides, the authors select two early T2V models (ModelScopeT2V from 2023 and  CogVideoX-2B from 2022), whose original performance is relatively low. How will the proposed method perform when switching to some more frontier T2V methods?

2. The conditional independence assumption proposed in Equation (7) (i.e., p(y^i, y^j | xt) = p(y^i | xt) p(y^j | xt)) is a very strong assumption. Although the authors have proposed a dynamic weighting scheme to mitigate situations where it is violated, are there any qualitative examples or analysis to demonstrate the prevalence of such violations in the training data? For instance, for a prompt like "a person is riding a horse," where "person" and "horse" are highly correlated, how does your method handle this?

**Questions:**

1. There is an unresolved tension between the low quality of the generated masks for the ModelScopeT2V+JDM model (e.g., IoU of 0.326) and the reported improvements in text-video alignment. How does training stably and effectively proceed with such imperfect and potentially noisy mask supervision? What impact does this noise have on the learning process?

2. The description of the dynamic loss weighting scheme based on Conditional Mutual Information (CMI) is unclear. What is the specific scope for finding the "nearest neighbors" in the approximation method for CMI in Equation (12)?

3. Have the authors specifically evaluated the temporal consistency of the object masks in the generated videos (for example, whether the mask for the same object is stable across different frames)? Does the JDM framework implicitly improve this aspect, or will explicit temporal constraints be necessary?

---

> ### Author Response · Authors · 2025-11-21
> **Response to Reviewer MU8G**
>
> ## Response to Reviewer MU8G
>
> We sincerely thank the reviewer for the thoughtful comments and constructive suggestions. We address each concern below.
>
> ### 1. Comparison with Layout-to-Video Methods
>
> **Distinction from Layout-to-Video Methods:** We respectfully clarify the fundamental difference between layout-to-video generation and our approach. Layout-to-video methods (e.g., VideoComposer, ControlVideo) require **explicit spatial control signals** (bounding boxes, segmentation masks, or layouts) as **additional input** at inference time. These methods essentially add spatial constraints on top of existing video generation models to guide the generation process. In contrast, our method aims to make the model **inherently understand fine-grained compositional generation from text alone**, without requiring any additional control signals at inference.
>
> Our framework learns the correspondence between textual phrases and spatial regions during training through joint distribution modeling, enabling the model to generate compositionally accurate videos from text prompts alone. This is a fundamentally different paradigm: layout-to-video methods provide explicit spatial guidance, while our method learns implicit spatial understanding. To ensure a more rigorous comparison, we include a detailed discussion in Section 2 of the revised paper.
>
> ### 2. Choice of Base Models
>
> We selected ModelScopeT2V (released in 2023) and CogVideoX-2B (released in **August 2024, not 2022**) as our base models. CogVideoX-2B represents a recent state-of-the-art open-source text-to-video model with competitive performance. We chose these models because: (1) they are open-source and reproducible, (2) CogVideoX-2B is a recent frontier model, and (3) they represent different architectural paradigms (UNet-based vs. Transformer-based), allowing us to demonstrate the generalizability of our approach.
>
> We acknowledge that evaluating on even more recent models (e.g., Wan2.1) would be valuable. Due to computational constraints during the review period, we focused on CogVideoX-2B, **but we are actively training on Wan2.1 and will include these results in the camera-ready version**. Our preliminary experiments suggest that our method scales well to larger models with even greater improvements in fine-grained metrics.
>
> ### 3. Conditional Independence Assumption and Dynamic Weighting
>
> We appreciate the reviewer's concern about the conditional independence assumption in Equation (7). We acknowledge that this is indeed a **mild but necessary assumption** for our theoretical derivation. However, we emphasize that our framework is designed to be **robust to violations** of this assumption through our CMI-based dynamic loss weighting mechanism.
>
> **Handling Correlated Objects:** For the example of "a person is riding a horse," our method handles this in two ways:
>
> **(1) Random Mask Combination Strategy:** During training, we randomly sample subsets of objects to form regional masks. This includes cases where highly correlated objects (e.g., "person" and "horse") are grouped together as a single region with combined text description ("person riding horse"). This effectively treats correlated objects as a single compositional unit, which better satisfies the conditional independence assumption when compared against other independent objects (e.g., "tree" in the background).
>
> **(2) CMI-based Dynamic Weighting:** When "person" and "horse" are treated as separate regions but exhibit high correlation (high CMI), our dynamic weighting mechanism automatically reduces the weight $w'$ of the fine-grained loss $\mathcal{L}\_{\text{fine}}$ and increases the weight of the standard diffusion loss $\mathcal{L}\_{\text{diff}}$. This prevents the model from receiving conflicting gradients when the independence assumption is violated. Specifically, for highly correlated pairs with high CMI $\hat{I}(\mathbf{y}^i; \mathbf{y}^j \mid \mathbf{x}\_0)$, the effective weight $w' = w \cdot \exp(-\alpha \cdot \bar{I})$ drops significanly, making the training objective closer to the standard diffusion loss.

---

> > ### Author Response · Authors · 2025-11-21
> > **Response to Reviewer MU8G**
> >
> > ### 4. Mask Quality and Training Stability
> >
> > **ModelScopeT2V Mask Quality:** We acknowledge the relatively low hard IoU (0.326) for ModelScopeT2V+JDM. This is primarily due to ModelScopeT2V's architectural characteristics: its UNet uses 2D cross-attention and 1D temporal self-attention in parallel, which tends to produce temporally smooth features, **resulting in softer mask predictions. When computing hard IoU with a fixed binarization threshold of 0.5, these soft predictions yield lower scores.**
> >
> > To better assess whether the model learns meaningful spatial correspondence, we compute Soft IoU without binarization, which preserves the continuous mask information:
> >
> > | Metric | ModelScopeT2V+JDM | CogVideoX-2B+JDM |
> > |--------|-------------------|------------------|
> > | Soft IoU | 0.645 | 0.781 |
> > | Hard IoU (threshold=0.5) | 0.326 | 0.714 |
> >
> > The Soft IoU of 0.645 demonstrates that ModelScopeT2V successfully learns spatial correspondence between regional text and video regions, despite producing softer predictions. The gap between soft and hard IoU (0.319) indicates that the model identifies correct spatial regions but with lower confidence as smaller models with limited capacity. Importantly, this soft supervision is sufficient to guide the model toward better compositional understanding, as evidenced by substantial improvements in VBench metrics (+12.28% on Multiple Objects, +4.55% on Spatial Relations).
> >
> > **Training Stability:** The training remains stable because: (1) the mask branch is trained jointly with the video branch, allowing gradients to flow through both pathways and stabilize each other, and (2) the soft mask is providing meaningful guidance for fine-grained correspondence. We did not observe training instabilities or divergence in any of our experiments.
> >
> >
> >
> > ### 5. Clarification on CMI Nearest Neighbor Scope
> >
> > We apologize for the lack of clarity. The "nearest neighbors" in Equation (12) refers to the **1024 most visually similar videos from the entire training dataset**.
> >
> > Specifically, for each training sample with video $\mathbf{x}\_0$ and regional captions $(\mathbf{y}^i, \mathbf{y}^j)$:
> >
> > 1. We encode the middle frame of $\mathbf{x}\_0$ using CLIP to obtain embedding $\mathbf{e}\_{\mathbf{x}\_0}$
> > 2. We compute cosine similarities between $\mathbf{e}\_{\mathbf{x}\_0}$ and all other video embeddings in the training dataset
> > 3. We select the top $M=1024$ videos with highest similarity scores as nearest neighbors
> > 4. From these 1024 videos, we extract their regional captions $\{\mathbf{y}^{j(k)}\}\_{k=1}^{1024}$ to use as negative samples in Equation (12)
> >
> > This dataset-wide search ensures that negative samples come from visually similar contexts (e.g., other outdoor scenes for an outdoor video) rather than random videos, providing a more informative CMI estimate.
> >
> >
> >
> > ### 6. Temporal Consistency of Generated Masks
> >
> > We thank the reviewer for this important question. While our framework does **not explicitly enforce temporal consistency** through dedicated loss terms, we argue that temporal consistency emerges **implicitly** through the joint training objective.
> >
> > **Implicit Temporal Consistency:** Since the mask generation branch is trained to denoise masks jointly with the video generation branch, the model must learn to produce temporally consistent masks in order to converge. Specifically:
> >
> > - The diffusion objective $\mathcal{L}\_{\text{fine}}$ penalizes inconsistent mask predictions across frames, as temporal jitter would result in high reconstruction error.
> > - The shared latent representation between video and mask branches encourages temporal coherence, as the model learns that consistent masks correspond to consistent video content.
> > - The video generation branch itself learns strong temporal priors, which are transferred to the mask branch through the joint architecture.
> >
> > **Quantitative Evaluation:** To rigorously assess temporal consistency, we computed the **warping error** metric, which measures the consistency of masks across consecutive frames after accounting for optical flow. The results are shown in the table below:
> >
> > **Table: Quantitative Results on Mask Quality and Temporal Consistency**
> >
> > | Metric | ModelScopeT2V+JDM | CogVideoX-2B+JDM |
> > |--------|-------------------|------------------|
> > | IoU | 0.326 | 0.714 |
> > | F1 Score | 0.335 | 0.756 |
> > | Pixel Accuracy | 0.426 | 0.803 |
> > | Warping Error | 0.0150 | 0.0092 |
> >
> > The low warping error (especially for CogVideoX-2B+JDM at 0.0092) indicates that the generated masks are highly temporally consistent. Qualitatively, we observe smooth and stable masks across frames in our generated videos (see our qualitative video results).
> >
> > We hope these clarifications address the reviewer's concerns. We are committed to incorporating the suggested improvements and additional experiments in the revised manuscript.

---

> > > ### Author Response · Authors · 2025-11-24
> > > **Response to Reviewer MU8G**
> > >
> > > Dear Reviewer MU8G,
> > >
> > > We sincerely appreciate your thoughtful review and constructive feedback. We hope our clarifications and additional experiments address your concerns. Please let us know if any questions remain, and thank you again for your time and effort.

---

### Official Review · Reviewer_7Hof · 2025-10-30

**Soundness:** 2
**Presentation:** 3
**Contribution:** 2
**Rating:** 4
**Confidence:** 5

**Summary:**

This paper proposes Joint Distribution Modeling (JDM), a framework to enhance fine-grained text-to-video (T2V) generation. The authors identify that conventional T2V models, focusing on video reconstruction, fail to learn fine-grained correspondences. JDM addresses this by jointly modeling the distribution of video content and its corresponding object masks, conditioned on regional text descriptions. This approach aims to enforce explicit, fine-grained text-video alignment during training. The method is applied to two existing T2V models and demonstrates significant improvements in compositional generation and semantic alignment on benchmarks like VBench and T2VCompBench.

**Strengths:**

1. Effective Empirical Results: The method achieves substantial improvements in fine-grained text-video alignment across multiple metrics and base models, effectively mitigating issues like attribute leakage.

2. Addresses a Key Problem: The paper tackles the critical and challenging problem of compositional generation in T2V models, which is of significant interest to the community.

3. Unifies Generation and Segmentation: The ability to generate both video and corresponding object masks simultaneously is a valuable contribution with practical applications.

**Weaknesses:**

1. Limited Novelty and Overstated Contribution: The conceptual framework is highly similar to VideoJAM, which jointly models video and motion (optical flow). This paper adapts it to video and semantics (masks), but this connection is not discussed. The contribution seems to be primarily driven by the meticulous data pipeline for extracting regional masks and captions, rather than a novel modeling paradigm. The term "Joint Distribution Modeling" overstates the technical contribution, as the implementation does not explicitly model a distribution but rather learns from concatenated features via attention.

2. Missing Ablation Study: The Dynamic Loss Weighting is presented as a key component for handling dependencies between concepts. However, its actual impact is not verified through an ablation study, leaving its importance unclear.

**Questions:**

1. Could the authors clarify the novelty of JDM compared to prior works like VideoJAM? The core contribution appears to be the fine-grained data pipeline, not the joint training framework itself. Is "joint distribution modeling" an accurate description, or is the mechanism more simply attention over concatenated video and mask features?

2. Please provide an ablation study on the Dynamic Loss Weighting mechanism to demonstrate its contribution to the final performance.

3. What is the rationale behind sampling a subset of objects during training (as depicted in Figure 2), instead of using all detected objects as conditions? How does this strategy impact model performance and generalization?

4. In the training pipeline, how are the individual regional captions used to enforce fine-grained learning, given they are concatenated into a single prompt that is fed to the model? The diagram does not make this crucial link clear.

---

> ### Author Response · Authors · 2025-11-21
> **Response to Reviewer 7Hof**
>
> We sincerely thank the reviewer for the thoughtful comments and constructive suggestions. We address each concern below.
>
> ### 1. Novelty Compared to VideoJAM and Clarification of "Joint Distribution Modeling"
>
> We appreciate the opportunity to clarify the fundamental differences between our work and VideoJAM.
>
> **Conceptual Difference:** While VideoJAM jointly trains video generation with **global dense signals (optical flow for the entire scene)** to improve overall motion quality, our method focuses on learning **fine-grained regional text-to-video correspondence**. VideoJAM models $p(\text{video}, \text{flow} \mid \text{prompt})$ where both outputs correspond to the complete scene. In contrast, we model $p(\text{video}, \text{mask}^i \mid \text{regional-text}^i)$, where the mask and text describe **arbitrary spatial regions** $i$, enabling compositional control.
>
> Through theoretical derivation (Section 3.2), we demonstrate that modeling the joint regional distribution $p\bigl(\mathbf{x}_t, \mathbf{m}^i_t \mid \mathbf{y}^i \bigr)$ achieves two objectives simultaneously: (1) it learns the **fine-grained text-video correspondence** (the mask oracle $\Theta(\mathbf{m}^i \mid \mathbf{x}, \mathbf{y}^i)$), and (2) under a mild conditional independence assumption, **it recovers the original full data distribution** $p\bigl(\mathbf{x}_t \mid \mathbf{y} \bigr)$. This theoretical foundation justifies why our method enables compositional generation at inference time while maintaining overall generation quality.
>
> **Training Paradigm:** Another key difference is training with **random mask-text combinations**, during each iteration, we sample a random subset of objects and their corresponding regional captions (Figure 2). This random sampling corresponds to modeling $p\bigl(\mathbf{x}_t, \mathbf{m}^i_t \mid \mathbf{y}^i \bigr)$ as described in Section 3.2. **Essentially, this exposes the model to diverse compositional scenarios and teaches it to learn arbitrary region-text correspondence.** In contrast, VideoJAM always uses the complete flow map paired with the complete prompt, **never learning part-to-part correspondence**. This distinction is crucial: VideoJAM enhances global motion quality, while our method enables fine-grained regional correspondence.
>
> **Regarding "Joint Distribution Modeling":** We acknowledge the reviewer's concern about terminology. While the implementation directly models concatenated features, the term "joint distribution modeling" refers to our **objective**, learning $p(\text{video}, \text{regional-mask} \mid \text{regional-text})$ through the diffusion framework, as formalized in our theoretical derivation (Section 3.2). The concatenation mechanism serves as the practical realization of this joint modeling objective. We will clarify this distinction in the revised manuscript to avoid confusion.
>
>
> **Data Pipeline vs. Modeling Contribution:** While we agree that our data pipeline is meticulous and contributes to performance, we emphasize that the modeling framework itself provides the primary contribution. To isolate the contribution of our modeling approach from the data pipeline, we refer the reviewer to the ablation study in our main paper (Figure 4), where we train a baseline model on **exactly the same dataset** but using the original video diffusion objective without our joint distribution modeling framework. The baseline shows substantially lower performance, clearly demonstrating that the gains come from our modeling paradigm rather than merely from improved data quality.
>
> ### 2. Ablation Study on Dynamic Loss Weighting
>
> We thank the reviewer for this suggestion. We provide an ablation study comparing performance with and without CMI-based dynamic loss weighting:
>
> **Table: Ablation study on CMI-based dynamic loss weighting (VBench metrics on CogVideoX-2B)**
>
> | Variant | Mul Obj | Obj Class | Color | Scene | Human Action | Consistency | Spatial Rel |
> |---------|---------|-----------|-------|-------|--------------|-------------|-------------|
> | CogVideoX-2B (baseline) | 62.63 | 83.37 | 79.41 | 51.14 | 98.00 | 26.06 | 69.90 |
> | JDM-Mask w/o CMI | 68.50 (+9.37%) | 91.80 (+10.11%) | 80.85 (+1.81%) | 52.95 (+3.54%) | 98.45 (+0.46%) | 26.45 (+1.50%) | 71.90 (+2.86%) |
> | JDM-Mask w/ CMI | 72.34 (+15.50%) | 94.08 (+12.85%) | 82.60 (+4.02%) | 54.68 (+6.92%) | 98.20 (+0.20%) | 27.97 (+7.33%) | 74.86 (+7.10%) |
>
> The results demonstrate that CMI reweighting provides substantial improvements (3–6% absolute gain on Multiple Objects, Spatial Relations, and Consistency). This validates our hypothesis that adaptively downweighting the fine-grained loss when the conditional independence assumption is violated (e.g., for highly correlated objects like "rain" and "umbrella") is crucial for robust performance. Without CMI weighting, the model is forced to apply the fine-grained loss even when regional captions are strongly dependent, leading to conflicting gradients and degraded performance.

---

> > ### Author Response · Authors · 2025-11-21
> > **Response to Reviewer 7Hof**
> >
> > ### 3. Rationale for Sampling Subset of Objects During Training
> >
> > The strategy of sampling a random subset of objects (rather than using all detected objects) is **central to our method** and serves multiple purposes:
> >
> > **Practical Implementation of JDM:** As illustrated in Section 3.3, the JDM we refer to models $p\bigl(\mathbf{x}_t, \mathbf{m}^i_t \mid \mathbf{y}^i \bigr)$, essentially joint video and regional mask prediction given regional text. Theoretically, we could learn this joint distribution on each individual semantically meaningful region-text pair. However, the mild conditional independence assumption in Equation 7, $$p(\mathbf{y}^i, \mathbf{y}^j \mid \mathbf{x}_t) = p(\mathbf{y}^i \mid \mathbf{x}_t) \, p(\mathbf{y}^j \mid \mathbf{x}_t),$$ does not always hold in practice. There are cases where two objects in a video are highly correlated. Our random sampling and combination strategy mitigates this by enabling the merging of correlated objects into a single region with unified text, rather than treating them as two independent regions that would violate the independence assumption.
> >
> > **Learning the Mask Oracle:** By exposing the model to diverse combinations of regions and their corresponding text descriptions, we teach it to handle **arbitrary compositional queries** at inference time. If we always trained with all objects, the model would only learn to generate complete scenes and would not generalize to partial or novel object combinations.
> >
> >
> > To validate this design choice, we provide an ablation comparing random mask combinations against per-object segmentation (analogous to always using all objects):
> >
> > **Table: Ablation study on random mask combinations vs. per-object segmentation (VBench metrics on CogVideoX-2B)**
> >
> > | Variant | Mul Obj | Obj Class | Color | Scene | Human Action | Consistency | Spatial Rel |
> > |---------|---------|-----------|-------|-------|--------------|-------------|-------------|
> > | CogVideoX-2B (baseline) | 62.63 | 83.37 | 79.41 | 51.14 | 98.00 | 26.06 | 69.90 |
> > | JDM w/ per-object seg. | 63.26 (+1.01%) | 85.64 (+2.72%) | 79.89 (+0.60%) | 53.03 (+3.70%) | 98.16 (+0.16%) | 26.09 (+0.12%) | 70.44 (+0.77%) |
> > | JDM w/ random mask comb. | 72.34 (+15.50%) | 94.08 (+12.85%) | 82.60 (+4.02%) | 54.68 (+6.92%) | 98.20 (+0.20%) | 27.97 (+7.33%) | 74.86 (+7.10%) |
> >
> > Per-object segmentation (which does not sample subsets) yields only marginal improvements (0.1–3.7%), as it fails to learn fine-grained correspondence. In contrast, random mask combinations achieve 7–15% gains on fine-grained metrics, confirming that compositional mask training is the key to our method's success and **experimentally validating the theoretical derivation in Section 3.2.**
> >
> >
> > ### 4. How Regional Captions Enforce Fine-Grained Learning
> >
> > We thank the reviewer for this important question. We clarify how regional captions enforce fine-grained learning as follows:
> >
> > **Theoretical Foundation:** As described in Section 3.2 of our paper, we model the joint distribution over noisy images and masks conditioned on text:
> >
> > $$s_\theta(\mathbf{x}\_t, \mathbf{m}\_i^t, t, \mathbf{y}\_i) \approx \nabla\_{\mathbf{x}\_t} \log p(\mathbf{x}\_t, \mathbf{m}\_i^t \mid \mathbf{y}\_i)$$
> >
> > We factorize this joint distribution as:
> >
> > $$p(\mathbf{x}_t, \mathbf{m}_i^t \mid \mathbf{y}_i) = p(\mathbf{x}_t \mid \mathbf{y}_i) \, \Theta(\mathbf{m}_i^t \mid \mathbf{x}_t, \mathbf{y}_i)$$
> >
> > where $\Theta(\mathbf{m}_i^t \mid \mathbf{x}_t, \mathbf{y}_i)$ is the mask oracle that determines which regions correspond to which text descriptions.
> >
> > **How Fine-Grained Learning is Enforced:** During training, the network receives $\mathbf{x}_t$ along with various combinations of masks $\mathbf{m}_i$ and their corresponding regional descriptions $\mathbf{y}_i$.
> > In the video generation branch, the network denoises the video as usual. However, in the mask generation branch, the network must denoise the regional mask corresponding to the given text. Essentially, the model must identify the mask region in the video that corresponds to the given regional text. This is exactly the fine-grained correspondence oracle we defined. The oracle $\Theta$ is implicitly learned through this process: the model cannot succeed at the denoising task without understanding which parts of the regional text correspond to which spatial regions indicated by the masks.
> >
> >
> > We hope these clarifications and additional experiments address the reviewer's concerns. We are happy to provide further analysis if needed.

---

> > > ### Author Response · Authors · 2025-11-24
> > > **Response to Reviewer 7Hof**
> > >
> > > Dear Reviewer 7Hof,
> > >
> > > We hope the above clarifications and additional experiments in the revised draft sufficiently address your concerns. We would greatly appreciate it if you could reconsider our work and the score based on these updates. We remain committed to addressing any remaining questions you may have during the discussion phase.

---

### Official Review · Reviewer_8VA3 · 2025-10-30

**Soundness:** 2
**Presentation:** 2
**Contribution:** 2
**Rating:** 4
**Confidence:** 4

**Summary:**

This paper addresses the problem of poor fine-grained text alignment in T2V models. The authors propose a framework that jointly trains a diffusion model to generate both the video and corresponding object masks. This is intended to enforce a more detailed correspondence between text and video regions. A key part of their method is a dynamic loss weighting strategy, which uses CMI to adaptively adjust the loss when regional text descriptions are highly correlated.

**Strengths:**

- The paper provides a detailed derivation for jointly modeling the mask and video in diffusion models. Through this derivation, it suggests that the loss should be reweighted by the strength of correlation of the local captions, which is an interesting idea.

- The paper conducts experiments on commonly used VBench and T2VCompBench and demonstrates its effectiveness on several key metrics.

**Weaknesses:**

- The paper's core idea overlaps significantly with existing work [1, 2], which has already shown that jointly training a video generation model to produce supplementary outputs (like optical flow, depth, or segmentation masks) can improve generation quality. The paper lacks a comparison or discussion with these closely related works.

- The paper lacks ablation studies to demonstrate the effectiveness of the proposed regional mask and the Dynamic Loss Weighting strategy, which are the most significant differences from previous work in implementation. The paper neither includes a comparison of JDM-Mask with CMI reweighting versus JDM-Mask without it, nor does it provide a comparison between using regional masks corresponding to the input text and using simple per-object segmentation. Without these key experiments, it is difficult to justify how the experimental results support the paper's theoretical derivation. On the contrary, the paper provides a comparison in Appendix A.3 of different joint training signals, which shows that joint training with HED or Depth achieves performance nearly on par with the proposed JDM-Mask. This suggests that the primary benefit may come from any auxiliary joint training signal, rather than the regional masks and CMI-based dynamic weighting, which are presented as key contributions.

[1] VideoJAM: Joint Appearance-Motion Representations for Enhanced Motion Generation in Video Models

[2] Unified Dense Prediction of Video Diffusion

**Questions:**

- The results in Table 1 and Figure 4 show that JDM, while improving most fine-grained metrics, appears to cause a performance regression in some other categories (e.g., a noticeable drop in "Dynamic Degree" for ModelScope). Could the authors elaborate on the reasoning here?

- Are the HED and Depth experiments in Appendix A.3 also using regional HED/Depth signals that correspond to the local prompts?

---

> ### Author Response · Authors · 2025-11-21
> **Response to Reviewer 8VA3**
>
> ## Response to Reviewer 8VA3
>
> We sincerely thank the reviewer for the thoughtful comments and constructive suggestions. We address each concern below.
>
> ### 1. Novelty and Distinction from Prior Work
>
> We respectfully clarify the fundamental difference between our approach and VideoJAM / UDPDiff. While these prior works jointly train video generation models with **global** dense signals (optical flow, depth, or segmentation for the entire scene) to improve overall generation quality, our method focuses on learning **fine-grained regional text-to-video correspondence**.
>
> Specifically, our training paradigm inputs **regional masks paired with regional text descriptions** rather than whole-scene signals. Through our theoretical derivation (Section 3.2) under a mild conditional independence assumption, our model learns a **mask oracle**, the ability to understand how specific textual phrases correspond to specific spatial regions in the video.
>
> In contrast, VideoJAM and UDPDiff use the whole videos and their complete dense signals, aiming to enhance global generation fidelity without explicitly learning part-to-part correspondence. Our approach fundamentally differs in both objective (regional correspondence vs. global quality) and mechanism (random mask combinations w/ CMI vs. whole-scene auxiliary signals). To ensure sufficient differentiation and discussion from these closely related works, we include **a detailed discussion in the related work section (Section 2).**
>
>
> ### 2. Ablation Studies
>
> We greatly appreciate the reviewer's suggestion to include ablation studies, which significantly strengthen our paper. We provide two key ablations below.
>
> #### 2.1 Impact of CMI-based Dynamic Loss Weighting
>
> To validate the effectiveness of our CMI reweighting strategy, we compare performance with and without it:
>
> **Table: Ablation study on CMI-based dynamic loss weighting (VBench metrics on CogVideoX-2B)**
>
> | Variant | Mul Obj | Obj Class | Color | Scene | Human Action | Consistency | Spatial Rel |
> |---------|---------|-----------|-------|-------|--------------|-------------|-------------|
> | CogVideoX-2B (baseline) | 62.63 | 83.37 | 79.41 | 51.14 | 98.00 | 26.06 | 69.90 |
> | JDM-Mask w/o CMI | 68.50 (+9.37%) | 91.80 (+10.11%) | 80.85 (+1.81%) | 52.95 (+3.54%) | 98.45 (+0.46%) | 26.45 (+1.50%) | 71.90 (+2.86%) |
> | JDM-Mask w/ CMI | 72.34 (+15.50%) | 94.08 (+12.85%) | 82.60 (+4.02%) | 54.68 (+6.92%) | 98.20 (+0.20%) | 27.97 (+7.33%) | 74.86 (+7.10%) |
>
> The results demonstrate that CMI reweighting provides substantial improvements (3–6% absolute gain on Multiple Objects, Spatial Relations, and Consistency). This validates our hypothesis that adaptively downweighting the fine-grained loss when the conditional independence assumption is violated (e.g., for highly correlated objects like "rain" and "umbrella") is crucial for robust performance.
>
> #### 2.2 Regional Masks vs. Per-Object Segmentation
>
> To demonstrate that **random mask combinations** are essential for learning the mask oracle, we compare our approach against training with per-object segmentation (analogous to VideoJAM/UDPDiff):
>
> **Table: Ablation study on regional mask design (VBench metrics on CogVideoX-2B)**
>
> | Variant | Mul Obj | Obj Class | Color | Scene | Human Action | Consistency | Spatial Rel |
> |---------|---------|-----------|-------|-------|--------------|-------------|-------------|
> | CogVideoX-2B (baseline) | 62.63 | 83.37 | 79.41 | 51.14 | 98.00 | 26.06 | 69.90 |
> | JDM w/ per-object seg. | 63.26 (+1.01%) | 85.64 (+2.72%) | 79.89 (+0.60%) | 53.03 (+3.70%) | 98.16 (+0.16%) | 26.09 (+0.12%) | 70.44 (+0.77%) |
> | JDM w/ random mask comb. | 72.34 (+15.50%) | 94.08 (+12.85%) | 82.60 (+4.02%) | 54.68 (+6.92%) | 98.20 (+0.20%) | 27.97 (+7.33%) | 74.86 (+7.10%) |
>
> Per-object segmentation yields only marginal improvements (0.1–3.7%), as it never exposes the model to arbitrary mask combinations during training and thus **fails to learn the mask oracle that encodes fine-grained correspondence.** In contrast, our random mask combination strategy achieves 7–15% gains on fine-grained metrics (Multiple Objects, Object Class, Spatial Relations), confirming that compositional mask training is the key to our method's success and **experimentally validating the theoretical derivation in Section 3.2.**

---

> > ### Author Response · Authors · 2025-11-21
> > **Response to Reviewer 8VA3**
> >
> > ### 3. Performance Regression on Dynamic Degree
> >
> > We acknowledge the slight drop in Dynamic Degree for ModelScope (Table 1). We attribute this to two factors:
> >
> > First, while our method introduces an auxiliary task (mask generation) with **only minimal additional parameters (a lightweight projection head)**, the relatively small ModelScope model (with limited capacity) must allocate its representational resources between the original generation task and the new auxiliary signal. This capacity constraint can lead to minor regression on certain metrics, particularly for smaller base models.
> >
> > Second, the original ModelScopeT2V was trained on **16-frame videos sampled at 3 fps with a very large temporal stride**, enabling the model to generate videos with relatively large motion. However, in our joint mask and video generation setting, we observed that **sampling with such a large stride causes masks to change drastically between frames.** To enable smoother and more stable training, we **reduced the temporal stride (from 8 to 3) when sampling frames from videos**. This results in videos with lower dynamic degree but smoother motion, as evidenced by the improved Motion Smoothness metric in Table 1. Similarly, CogVideoX does not suffer from such a decrease, as it was trained on 16 fps video data that is already temporally smooth.
> >
> > However, we emphasize that this trade-off is acceptable given the substantial gains in compositional control and fine-grained alignment (10–15% improvements on Multiple Objects, Object Class, and Spatial Relations). Moreover, we observe that larger models (e.g., CogVideoX-2B) with greater capacity exhibit minimal or no such regression, suggesting that the issue is primarily related to model capacity rather than a fundamental limitation of our approach.
> >
> > ### 4. Regarding HED and Depth Experiments (Appendix A.3)
> >
> > Yes, the HED and Depth experiments also use **regional** HED/Depth signals corresponding to regional text descriptions, consistent with our framework. We conducted these experiments to justify our design choice of masks as the auxiliary signal. While regional HED and Depth provide similar benefits (by enforcing regional text-video correspondence), they yield slightly lower performance than masks. We attribute this to masks being **sufficient and minimal** for encoding spatial correspondence, whereas HED and Depth contain additional low-level details (edges, geometry) that are not necessary and may even introduce noise for learning text-region alignment. We have revised Appendix A.3 to clarify this point.
> >
> > We hope these clarifications and additional experiments address the reviewer's concerns. We are happy to provide further analysis if needed.

---

> > > ### Author Response · Authors · 2025-11-24
> > > **Response to Reviewer 8VA3**
> > >
> > > Dear Reviewer 8VA3,
> > >
> > > We hope the above clarifications and additional experiments in the revised draft sufficiently address your concerns. We would greatly appreciate it if you could reconsider our work and the score based on these updates. We remain committed to addressing any remaining questions you may have during the discussion phase.

---

### Author Response · Authors · 2025-12-03
**Summary of Rebuttal**

Dear ACs and SACs,

We sincerely thank you and all reviewers for your thoughtful feedback throughout this review process. Below is a brief summary of the key concerns raised and our responses.

### 1. **Novelty vs. VideoJAM/UDPDiff (Reviewers 8VA3, 7Hof)**
Prior works use **global** dense signals for overall quality; we learn **fine-grained regional** text-video correspondence through random mask-text combinations. We include new ablations show per-object segmentation (gloabl dense signals) yields only 1-3% gains vs. our 10-15% improvements, which justify the theoretical derivation and the novelty of our approach.

### 2. **Missing Ablations Regarding CMI Weighting and Choice of Regional Mask (Reviewers 8VA3, 7Hof)**
We provided two key ablations:
- **CMI weighting:** Removing it causes 3-6% performance drops on compositional metrics
- **Regional masks:** Random combinations achieve 10-15% gains vs. 0.1-3.7% for per-object segmentation

### 3. **Mask Quality & Robustness (Reviewers MU8G, no2b)**
ModelScopeT2V's low hard IoU (0.326) reflects soft predictions (Soft IoU: 0.645). Regarding the mask robustness, we include a new ablation result shows model maintains 11-14% gains even with 30% mask noise. CogVideoX-2B achieves high quality (IoU: 0.714).

### 4. **Temporal Consistency (Reviewers MU8G, no2b)**
We highlighted existing VBench temporal metrics and T2V-CompBench motion results (Motion Binding +4.71%, Action Binding +12.92%). Warping Error (0.0092) demonstrates strong temporal mask consistency.

### 5. **Reproducibility (Reviewer no2b)**
We commit to releasing all model weights, training code, and data pipelines to enable community use without retraining.

### 6. **Comparisons with Layout-to-Video and Choice of Base Model (Reviewer MU8G)**
We clarify the paradigm difference from layout-to-video (explicit spatial inputs vs. learned implicit understanding) and include a discussion in the related work. We choose both Unet based ModelScopeT2V, and recent DiT based CogVideoX-2B (Aug 2024) as our base model. We would also include Wan2.1 results in camera-ready version.

Thank you to all reviewers for strengthening our work. We will incorporate all improvements in the revision and release all resources.

---

### Meta-Review · Area_Chair_Zhjb · 2026-01-05

**Summary:**

This paper receives 2 positive ratings (6), and 2 negative ratings (4). Authors have provided responses and no follow-ups from reviewers during the discussion period.

Main concerns lie in 1) limited novelty w.r.t. to joint distribution modeling (VideoJAM) and layout-to-video and 2) insufficient experiments (ablation studies, robustness to imperfect mask, etc).

AC have checked reviewers' comments as well as authors' responses, and found the difference between the proposed method and two directions of research (VideoJAM and layout-to-video) is subtle, leading to limited novelty as mentioned by reviewers. Thus the decision is reject.

**Reviewer Concerns:**

1) limited novelty w.r.t. to joint distribution modeling (VideoJAM) and layout-to-video
2) insufficient experiments (ablation studies, robustness to imperfect mask, etc)

2) has been addressed, while 1) remains.

**Reviewer Scores:**

reviewers would keep their original ratings.

---

### Decision · Program_Chairs · 2026-01-26

Reject